# Topology selectivity of a conformationally flexible precursor through selenium doping

Liangliang Cai [1], Tianhao Gao[1] & Andrew T. S. Wee [1] ✉

Conformational arrangements within nanostructures play a crucial role in shaping the overall configuration and determining the properties, for example in covalent/metal organic frameworks. In on-surface synthesis, conformational diversity often leads to uncontrollable or disordered structures. Therefore, the exploration of controlling and directing the conformational arrangements is significant in achieving desired nanoarchitectures. Herein, a conformationally flexible precursor 2,4,6-tris(3-bromophenyl)−1,3,5-triazine is employed, and a random phase consisting of $C_{3h}$ and $C_s$ conformers is firstly obtained after deposition of the precursor on Cu(111) at room temperature to 365 K. At low coverage (0.01 ML) selenium doping, we achieve the selectivity of the $C_{3h}$ conformer and improve the nanopore structural homogeneity. The ordered two-dimensional metal organic nanostructure can be fulfilled by selenium doping from room temperature to 365 K. The formation of the conformationally flexible precursor on Cu(111) is explored through the combination of high-resolution scanning tunneling microscopy and non-contact atomic force microscopy. The regulation of energy diagrams in the absence or presence of the Se atom is revealed by density functional theory calculations. These results can enrich the on-surface synthesis toolbox of conformationally flexible precursors, for the design of complex nanoarchitectures, and for future development of engineered nanomaterials.

On-surface synthesis has been investigated in the past decades owing to its synthetic accessibility of diverse nanostructures[1–4]. Various complex nanostructures have been achieved by smart design of precursors, choice of substrates, and delicate experimental parameters such as molecular concentration, electrical stimulation, thermal treatment, etc.[5–8]. Among diverse on-surface reactions, the Ullmann coupling has been a milestone and one of the most explored reactions in on-surface science, in which precursors undergo dehalogenation, organometallic (OM), and covalent interlinking effectively and efficiently[9–12]. To date, most on-surface synthesis explorations have been primarily concentrated in conformationally rigid precursors[13–18], with limited attention paid to conformationally flexible precursors.

Investigating on-surface reactions of conformationally flexible precursors is of significance due to the potential in forming complex functional nanomaterials with engineered structures and properties[19–22].

However, controllable synthesis of nanostructures from conformationally flexible precursors is challenging due to poor selectivity among various conformations. Some strategies have been demonstrated to control the molecular conformation, such as the template effect, annealing treatment, adsorbed bromine atoms, etc.[23–27]. However, transformation from random structure to two-dimensional (2D) crystalline structure with one specific precursor conformation (e.g., 2D nanopores) is hard to accomplish. There is literature on the regulation of metal or ion doping on-surface 2D networks[28,29], which could to some extent help us understand the doping effects, but specifically, selenium (Se) adatoms/clusters effect on surface 2D network has not yet been studied to our knowledge, and this topic is important in view of the increased interest in selenides.

In this work, the conformationally flexible precursor 2,4,6-tris(3-bromophenyl)−1,3,5-triazine (mTBPT) is adopted, in which the triazine

[1]Department of Physics, National University of Singapore, 2 Science Drive 3, Singapore 117542, Singapore. ✉e-mail: phyweets@nus.edu.sg

ring is equipped with three *meta*-bromophenyl groups. As shown in Fig. 1, conformers with $C_{3h}$ and $C_s$ symmetries can be distinguished by the rotation of the *m*-bromophenyl groups around the C−C σ-bonds to the central triazine ring. The conformationally flexible *m*TBPT becomes chiral upon 2D confinement. Hence, it is challenging to control the conformation selectivity and the fabricated nanoarchitectures. A random phase consisting of $C_{3h}$ and $C_s$ conformers with both left (L) and right (R) handed chirality is obtained after deposition of *m*TBPT on Cu(111) at room temperature (RT) to 365 K as expected. We have investigated the low coverage (0.01 ML) Se doping of conformationally flexible *m*TBPT on Cu(111) and achieved topology selectivity. The high selectivity of $C_{3h}$ conformer greatly improves the structural homogeneity, so that a monolayer-ordered 2D metal-organic framework (MOF) is achieved. Co-deposition of *m*TBPT and Se take effect at RT to 365 K, and the deposition sequence of *m*TBPT and Se does not make any difference. In addition, the random phase consisting of both $C_{3h}$ and $C_s$ conformers can be transformed to the ordered crystalline phase by 0.01 ML Se doping at RT. Annealing to

365 K accelerates the process from a kinetic point of view. Using a combination of high-resolution scanning tunneling microscopy/spectroscopy (STM/STS) and non-contact atomic force microscopy (nc-AFM) at 4 K, we have explored the controllable fabrication of the conformationally flexible precursor *m*TBPT on a Cu(111) substrate and achieved the high topology selectivity of *m*TBPT by Se doping. Density functional theory (DFT) calculations further reveals the regulation of energy diagrams in the absence or presence of Se and explain the transformation from the random phase to the ordered 2D MOF. These results are potential to enrich the on-surface synthesis toolbox of conformationally flexible precursors, encouraging the design of intricate nanoarchitectures.

## Results

### Intact conformers and the random phase before Se doping
Deposition of submonolayer amounts of *m*TBPT on cold Cu(111) held at 90 K results in the formation of isolated monomers and self-assembled domains with limited order as illustrated in Fig. 2a.

**Fig. 1 | 2,4,6-tris(3-bromophenyl)−1,3,5-triazine conformers with $C_s$ and $C_{3h}$ symmetries, right-handed and left-handed chirality.** Reaction pathway of *m*TBPT on Cu(111) toward random organometallics and crystalline 2D metal-organic nanostructures after selenium doping.

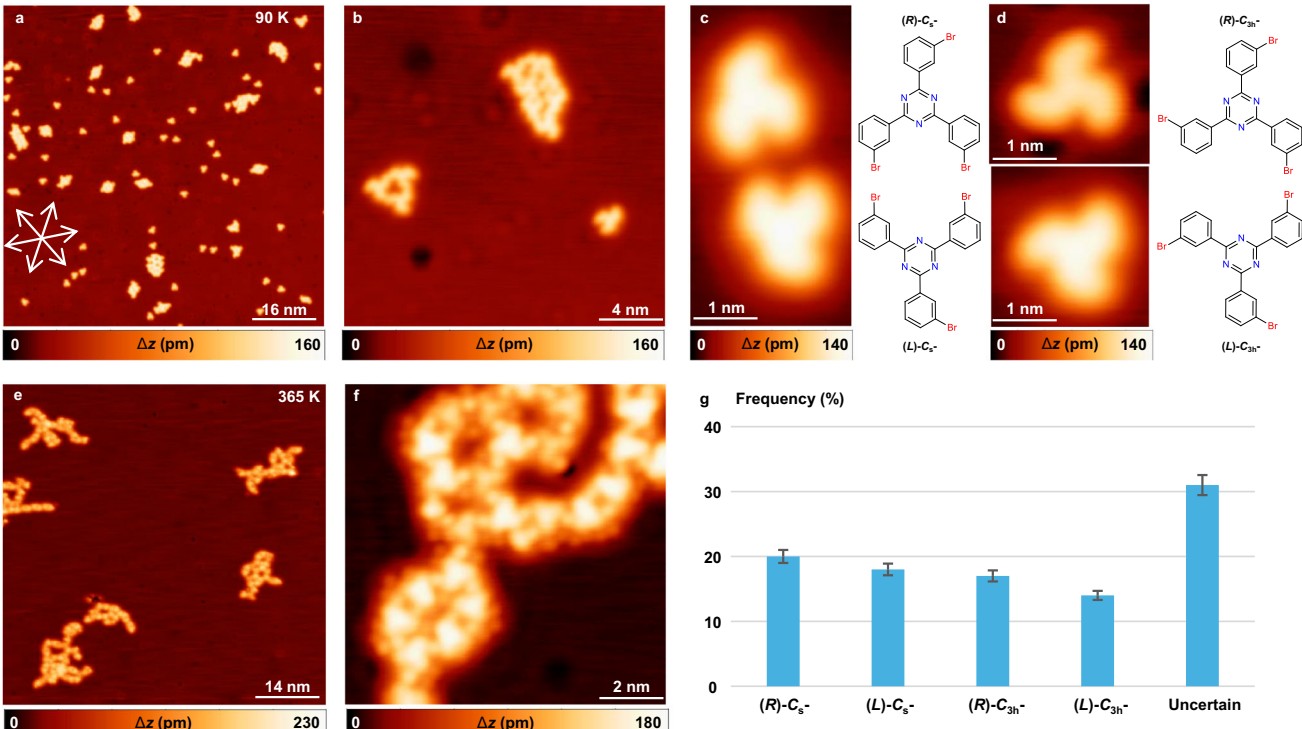

**Fig. 2 | Intact conformers and the random phase formed by *m*TBPT on Cu(111).** **a** Overview and **b** close-up STM images showing isolated monomers and self-assembled domains with limited order after deposition of *m*TBPT on Cu(111) at 90 K. The close-packed directions of Cu(111) are marked by white arrows. **c, d** High-resolution STM images indicating the intact conformers (*R*)-*C*_s-, (*L*)-*C*_s-, (*R*)-*C*_3h-, and (*L*)-*C*_3h-*m*TBPT together with their schematic models. **e** Overview and **f** close-up STM images after annealing to 365 K, showing the random phase of C−Cu−C organometallics formed by fully debrominated *m*TBPT precursors and Cu adatoms. **g** Statistical analysis of 1200 molecules from 20 images showing no obvious selectivity of *m*TBPT conformers on Cu(111) with error bars within 5%. STM parameters: constant current, **a**, **e**, **f** $U = 1$ V, $I = 100$ pA; **b**–**d** $U = -1$ V, $I = 100$ pA.

The close-up STM measurement in Fig. 2b indicates that the porous network is exclusively formed by intact 3-fold symmetric $C_{3h}$-*m*TBPT conformers, and the disordered domain consists mainly of intact non-3-fold symmetric $C_s$-*m*TBPT conformers. No specific preference was observed for different conformers including $C_s$ and $C_{3h}$ symmetries with (*L*) and (*R*) chirality of the *m*TBPT precursor. Four *m*TBPT conformers can be discerned in the high-resolution STM images in Fig. 2c, d, including (*R*)-$C_s$-, (*L*)-$C_s$-, (*R*)-$C_{3h}$-, and (*L*)-$C_{3h}$-*m*TBPT conformers.

After annealing the sample to 365 K, random structures were observed as depicted in Fig. 2e. Upon closer inspection in Fig. 2f, the *m*TBPT precursor was fully debrominated and no intact C−Br moiety was detected. There is no obvious selectivity of *m*TBPT conformers from a statistical analysis of 1200 molecules as shown in Fig. 2g, which explains the formation of the random C−Cu−C OM nanostructure. (More detailed discussion on theoretical calculations see below.) Different thermal treatments including deposition of the precursor at RT or higher temperature (365 K) did not show any selectivity towards a specific conformer (Supplementary Fig. 1), either.

**Ordered 2D organometallic networks through Se doping**
Se doping has proved to have an influence on self-assembly and crystallization behavior[30,31], so we explored the low coverage (0.01 ML) Se doping of the conformationally flexible precursor *m*TBPT on Cu(111) substrate and achieved high topology selectivity. The selectivity of $C_{3h}$ conformer was greatly improved, so that the ordered 2D MOF could be achieved after deposition of 0.28 ML *m*TBPT and 0.01 ML Se on Cu(111) at 365 K as shown in Fig. 3a. From the close-up STM image in Fig. 3b, (*L*) and (*R*) domains were formed, deviating 15 ± 3° from the [01̄1] direction of the substrate. From the statistical analysis of 1200 molecules (Fig. 3c), the selectivity of $C_{3h}$ conformers is quite high (close to 90%).

The conformation of some moieties at the periphery of the networks is hard to be determined, so we classify them into the uncertain part. No exclusive $C_s$ incorporated nanostructures (*i.e.* 1D OM chains) were observed. Furthermore, the random phase consisting of both $C_{3h}$ and $C_s$ conformers can be transformed into the ordered crystalline phase by Se doping at RT (Supplementary Fig. 2). Annealing to 365 K is not necessary for the transformation but can expedite the process. From Supplementary Fig. 2c, d, it is clear that the proportion of non-hexagon monomers in the 2D networks has increased at RT. This means thermal deposition (at 365 K) increases the proportion of hexagonal OM rings. At higher temperature, molecules have higher mobility on the surface, which is more conducive to the self-healing of 2D OM networks, and achieve a more thermodynamically stable structure. In view of kinetics, mild annealing for sufficient time (i.e. RT for 12 h) could also increase the proportion of hexagon pores in the formed network as illustrated in Supplementary Fig. 2e, f.

In the high-resolution STM image of (*L*) network (Fig. 3d), a green line is drawn from point A to point B, representing a line scan profile (Fig. 3e). The measured dimension of the *m*TBPT−Cu−*m*TBPT species is shown in panel (e), indicating the height difference between the neighboring *m*TBPT moieties. This will be further clarified in the nc-AFM (Fig. 4) images. The unit cell is indicated by a black dashed rhombus with $a = b = 22.8 \pm 0.8$ Å, $\alpha = 60 \pm 1°$. In the high-resolution image of the (*R*) domain (Fig. 3f), red and black triangles outline brighter (I) and dim (II) molecules, respectively, while blue circles contour the interlinked Cu adatoms. Normally the brighter and dim molecules are aligned alternatively. A dashed triangle frames a bright molecule which is supposed to be dim. We can take it as a "defect". The reason might be attributed to the incommensurate registry between the MOF and the underlying substrate lattice. The DFT-relaxed 2D MOF

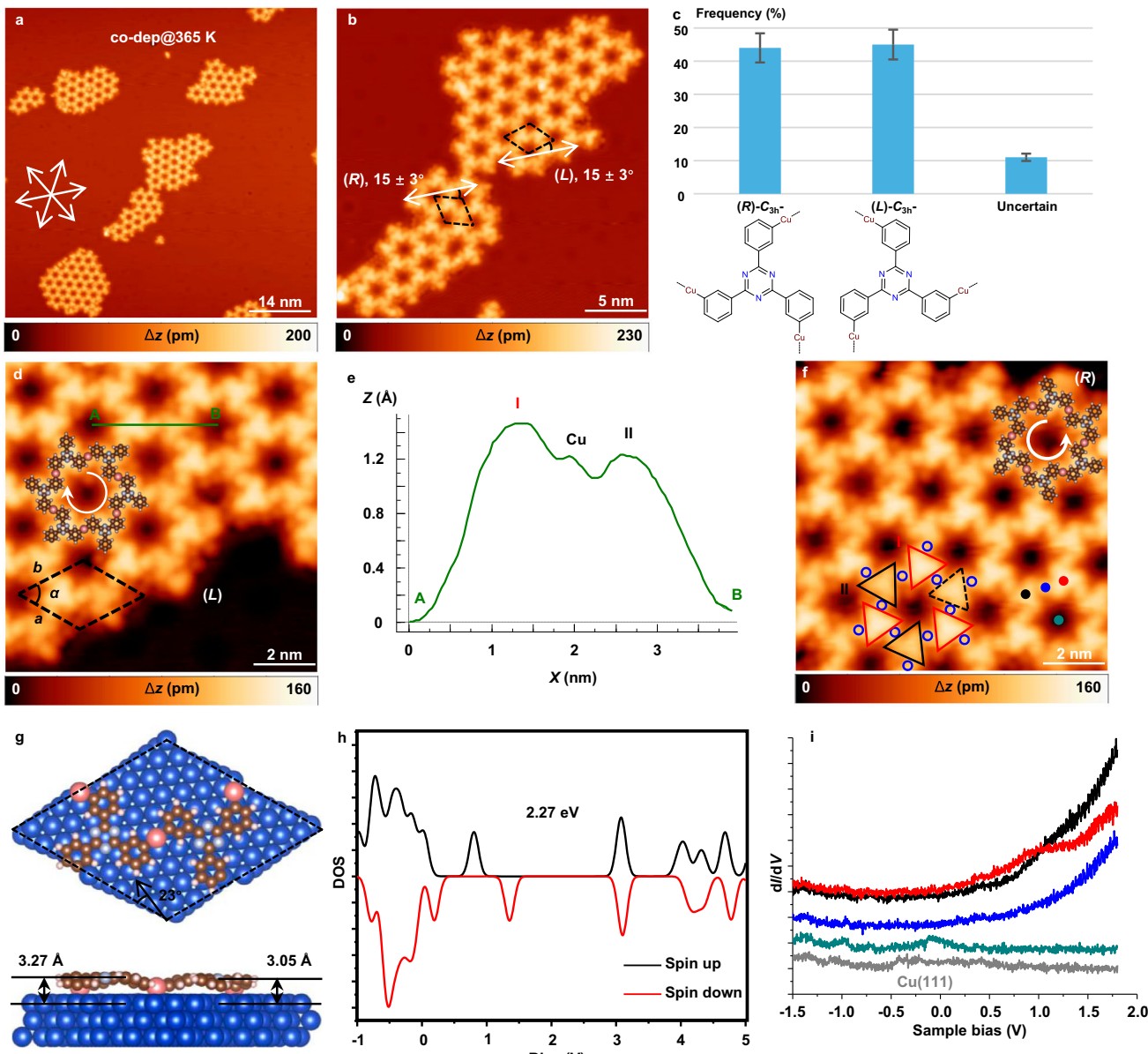

**Fig. 3 | Crystallization of 2D organometallic networks on Cu(111) after Se doping. a** Overview and **b** close-up STM images showing the ordered 2D organometallic networks fabricated by co-deposition of $m$TBPT and selenium on Cu(111) at 365 K. ($L$) and ($R$) domains of the formed networks, deviating 15 ± 3° from the [0$\bar{1}$1] direction of the substrate. **c** Statistical analysis of 1200 molecules showing high selectivity of $m$TBPT forming $C_{3h}$ moieties on Cu(111) after selenium doping with error bars within 5%. **d** High-resolution of ($L$) domain with molecular model partially fitted. The green line from A to B is a line scan profile with the measured dimension of the $m$TBPT−Cu−$m$TBPT species shown in (**e**). The unit cell is indicated by a black dashed rhombus. **f** High-resolution of ($R$) domain with molecular model partially fitted. Red and black triangles contour the brighter (I) and dim (II) molecules, and blue circles contour the interlinked Cu adatoms. The dashed triangle frames a

brighter molecule. **g** Top-view and side-view of the DFT-relaxed building block, showing different adsorption heights of the triazine cores. The unit cell is indicated by a black dashed rhombus, deviating 23° from the [0$\bar{1}$1] direction of the substrate. Brown, white, gray, blue, and pink balls represent C, H, N, Cu substrate, and Cu adatoms, respectively. **h** DFT calculated spin-polarized density of states of the ordered organometallic network in gas phase. Black and red lines represent the spin-up and spin-down contributions, respectively. **i** d$I$/d$V$ spectra recorded at different sites (green, pore; blue, Cu adatom; black and red, dim and brighter molecules) as indicated in (**f**). The tip is fully calibrated in bare Cu(111) (gray curve). STM parameters: constant current, **a, b, d, f** $U$ = 1 V, $I$ = 100 pA. Source data are provided as a Source data file.

on Cu(111) is depicted in Fig. 3g, with the unit cell indicated by a black dashed rhombus, deviating 23° from the [0$\bar{1}$1] direction of the substrate. The dimensions of the unit cell are calculated to be $a = b = 22.28$ Å, $\alpha = 60°$. The side view of the DFT-relaxed building block (−Cu−$m$TBPT−Cu−$m$TBPT−Cu−) in Fig. 3g clearly exhibits the height difference of the triazine cores.

To investigate the mechanism of Se doping, we performed additional experiments with different coverage of Se and different ratios of

$m$TBPT to Se. All of these samples were thermally treated at 365 K for 2 h after co-deposition of $m$TBPT and Se.

When 0.005 ML Se and 0.28 ML $m$TBPT were mixed on Cu(111), there coexisted $C_s$ and $C_{3h}$ conformers, forming OM chains and networks as shown in Supplementary Fig. 3a, b. After increasing the coverage of Se to 0.01 ML (Supplementary Fig. 3c, d), only the $C_{3h}$ conformer was observed, leaving some small copper selenide domains adjacent. That is to say, the amount of Se is related to the stabilization

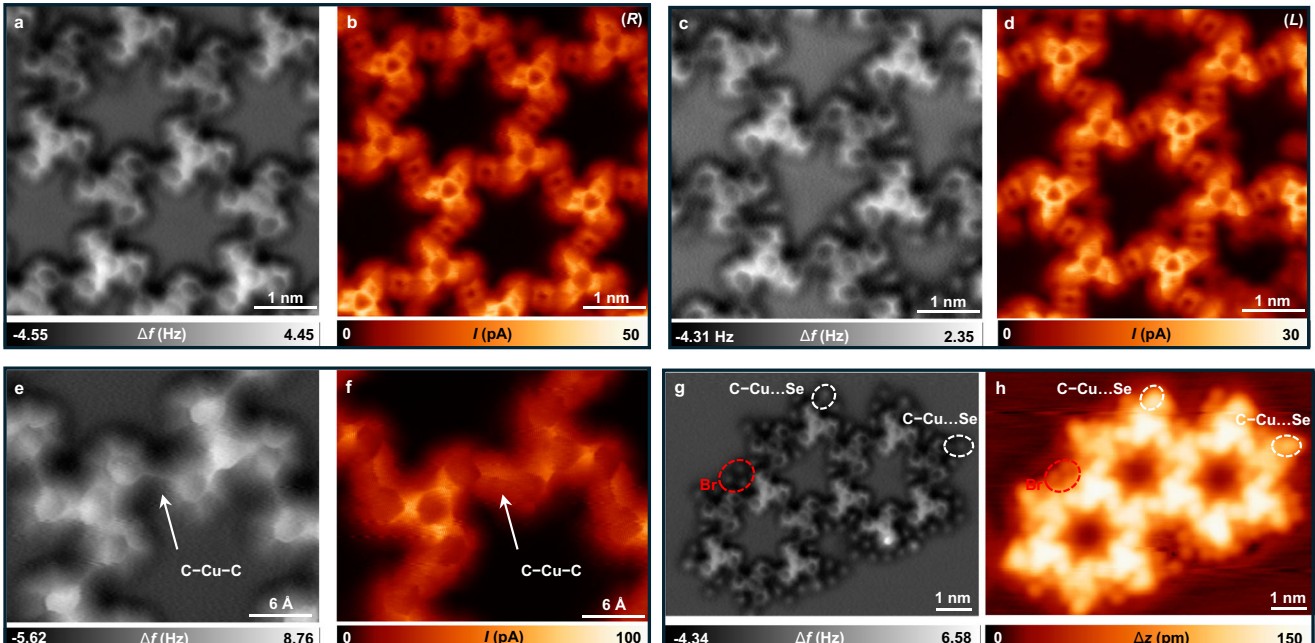

**Fig. 4 | The structure of 2D MOFs on Cu(111). a** The nc-AFM frequency shift image and **b** current image acquired simultaneously of the (R) domain. **c** The nc-AFM frequency shift image and **d** current image acquired simultaneously of the (L) domain. **e** High-resolution of nc-AFM frequency shift image and **f** current image acquired simultaneously indicating the C−Cu−C interlinking. **g** The nc-AFM frequency shift image showing the periphery interactions. **h** The corresponding current image acquired at the same position of panel (**g**). The C−Cu...Se interactions are indicated by white ellipses, and detached Br atoms by red ellipses, respectively. Scanning parameters: **a–g** constant-height mode using a CO-terminated tip at a bias voltage of 0 V; **h** constant current mode with $U = 1$ V, $I = 100$ pA. Images (**a–g**) were recorded at different tip offsets $\Delta z$ (**a, b** $\Delta z = -250$ pm; **c, d** $\Delta z = -240$ pm; **e–g** $\Delta z = -260$ pm) with respect to an STM set point ($U = 1$ V, $I = 100$ pA).

of the $C_{3h}$ conformer. The coverage of $m$TBPT was increased to 0.70 ML (Supplementary Fig. 3e, f), the network grows bigger, and the chain structures composed of the $C_s$ conformer was observed again due to insufficient Se. By increasing Se to 0.015 ML (Supplementary Fig. 3g, h), the $C_s$ conformer was observed less. Therefore, we hypothesize that the $C_{3h}$ conformer could be stabilized with the doped Se atoms. Considering the low diffusion barrier of Se atoms on Cu(111) (0.107 eV)[32], it is plausible that the Se atoms play an important role in regulating the precursor conformers and topology selectivity. More details about how the Se atoms influence the energy diagram are provided in the DFT discussion.

We also carried out STS measurements on Cu(111)-supported 2D MOFs at different sites as indicated in Fig. 3f. The d$I$/d$V$ spectrum on the Cu(111) substrate shows a small Shockley surface state at approximately −450 mV (gray curve in Fig. 3i)[33]. The green spectrum is measured at the center of one pore and reveals the confined surface state of Cu(111)[34–37]. Different pores are decorated with different numbers of residual Br atoms that interact with H atoms of the benzene ring. The STS measurements (Supplementary Fig. 4) show a stepwise upshift of the confined surface state related to the attachment of an increasing number of Br atoms (0–3) in the pores, which is consistent with Br acting as an electron acceptor leading to a $p$-doping effect[38,39]. The peak at about +1.0 V(red curve) of the d$I$/d$V$ spectrum was resolved in the unoccupied region, and we assign it to the conduction band edge (CBE).

To explore the origin of the state in the STS, we calculated the density of states (DOS) for the 2D OM network in gas phase by DFT. Figure 3h displays the spin-polarized DOS of the free-standing 2D OM network. Intriguingly, the asymmetric DOS for spin-up and spin-down components indicates a spin polarization of the 2D OM network, although the incorporated Cu atoms in the chains are not magnetically active, which is similar to our previous studies[40,41].

The comparison of the measured data on the surface (Fig. 3i) with the calculated results in gas phase (Fig. 3h) reveals the influence of the Cu(111) surface on the electronic properties of the 2D OM network. The metal surface transfers electrons to the network as an electron donor, which is accompanied by a shift of all the bands downward in energy by about 2 eV with respect to the Fermi level, and is similar to Au− and Ag−involved OM networks[39,42]. No more details were observed from d$I$/d$V$ spectra recorded across the $m$TBPT − Cu−$m$TBPT bridge, which could be attributed to the dominant contribution of the DOS from the copper substrate[43,44].

The corresponding projected DOS (PDOS) on the Cu, C, and N elements of the 2D OM network was also calculated to show individual contributions, where the Cu and C orbitals contribute to the highest occupied molecular orbitals (HOMO) and the C and N orbitals contribute to the lowest unoccupied molecular orbitals (LUMO) of the 2D OM network (Supplementary Fig. 5b).

## Nc-AFM characterization

CO-adsorbed tip nc-AFM[45] was next conducted at 4 K. The nc-AFM frequency shift images and corresponding current images of (R) (Fig. 4a, b) and (L) (Fig. 4c, d) domains clearly display the molecular buckling and height difference of the formed 2D metal-organic nanostructures. The high-resolution nc-AFM frequency shift image (Fig. 4e) and the simultaneously acquired current image (Fig. 4f) distinctly exhibit the C−Cu−C interlinking with bond length of 3.8 ± 0.1 Å from carbon to carbon, in agreement with the theoretical parameter 3.86 Å. The periphery of the network offers valuable clues that can shed light on the underlying mechanism of regulating crystallization through Se doping.

As illustrated in Fig. 4g, h, terminal interactions to the periphery molecules were indicated by white ellipses. Considering that there is no such interaction at the periphery molecules before Se doping

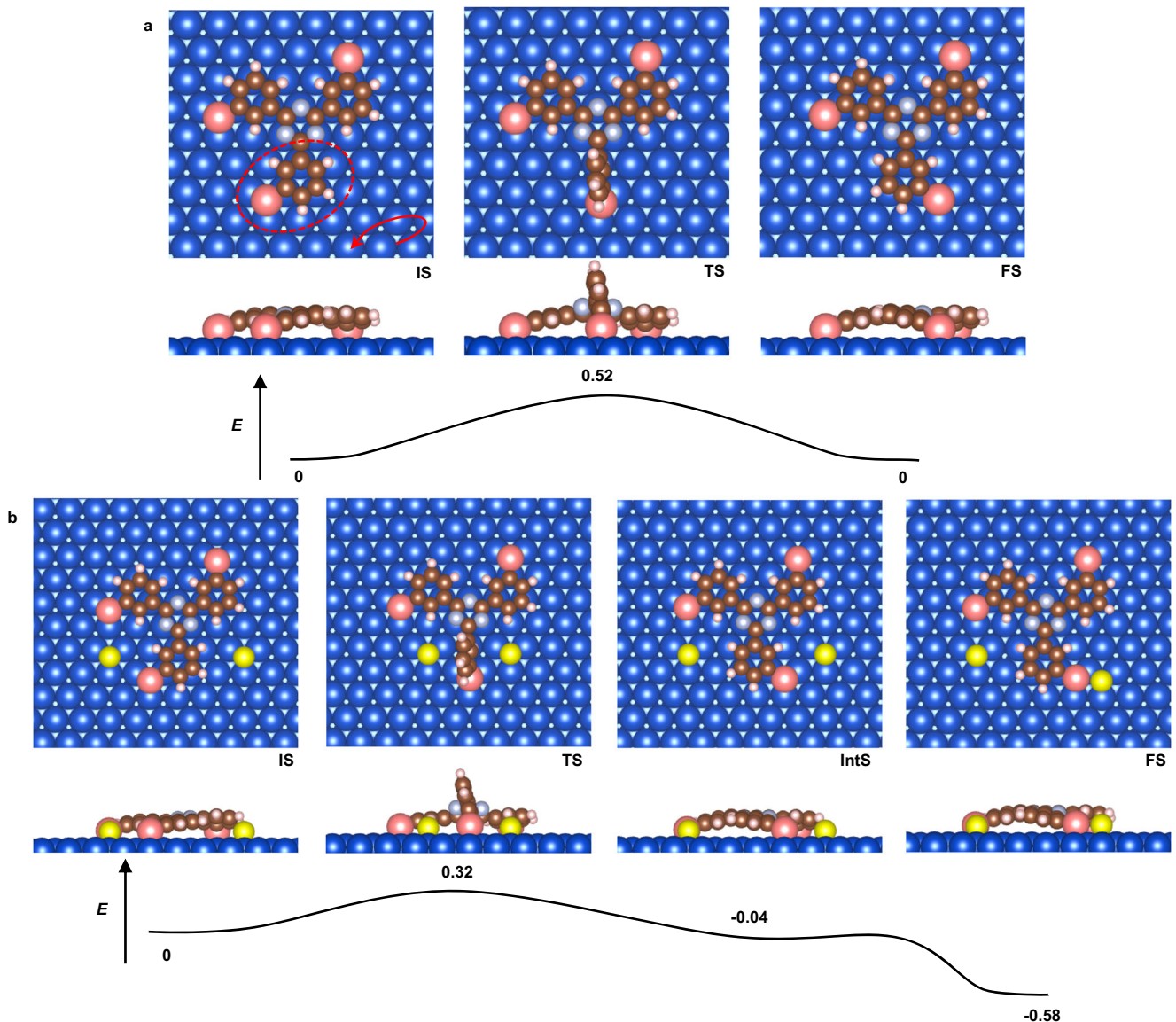

**Fig. 5 | Topology transformation pathway from $C_s$ to $C_{3h}$ in the absence or presence of Se atoms.** Energy diagrams for the transformation from $C_s$ to $C_{3h}$ **a** without and **b** with Se doping. The structural models are given for the initial, transition, intermediate, and final states, respectively. The energy scale is not linear. Energies are given in units of eV. Brown, white, gray, yellow, blue, and pink balls represent C, H, N, Se, Cu substrate, and Cu adatoms, respectively. Source data are provided as a Source data file.

(surface stabilized radicals as shown in Supplementary Fig. 2b), we infer that the interaction is attributed to Se. Combined with the high-resolution nc-AFM and STM images, we assign the terminal interactions (depicted by white ellipses) to C−Cu...Se interactions as shown in Supplementary Fig. 6e with a model superimposed. The bond length of the C−Cu...Se interaction equals to 4.2 ± 0.1 Å from carbon to Se, in agreement with the theoretical parameter 4.23 Å. This suggests that there is interaction between the Se atom and the C−Cu species.

The interaction region indicator (IRI) analysis on the C−Cu...Se moiety was also performed to investigate the intermolecular interactions. To illustrate the nature of interactions between Se adatoms and C−Cu species, we used a finite C−Cu moiety with one Se atom terminated as a model system in the IRI analysis. IRI is a real-space function that can clearly reveal both chemical bonds and weak interactions in chemical systems[46]. As illustrated in the IRI isosurface (Supplementary Fig. 7), both covalent bond and weak interaction regions are revealed by the blue and green isosurfaces, respectively, according to the standard coloring method. This thus demonstrates that there is

nonnegligible interaction between the Se adatom and the C−Cu species as evidenced by nc-AFM observations.

## Mechanism
To better understand the mechanism for the high topology selectivity of the specific $C_{3h}$ conformers out of a conformationally flexible precursor by Se doping, nudged elastic band (NEB) calculations in the absence or presence of Se have been employed to model the transformation between $C_s$−Cu and $C_{3h}$−Cu moieties on Cu(111). As shown in Fig. 5a, the energy barrier of $C_s$ transformed to $C_{3h}$ without Se was calculated to be 0.52 eV, which can be overcome at RT. Considering the similar energy of $C_s$ and $C_{3h}$ conformers, it is expected that the random OMs were formed with no obvious selectivity.

After Se doping, the Se atoms locate beside the phenyl lobe which is about to rotate (IS in Fig. 5b). During the rotation, the adjacent Se atoms help to decouple the organic backbone from the surface and prevents hybridization, which would facilitate rotation as the molecule-substrate interactions are weaker. The rotation barrier from

initial state via transition state to intermediate state is calculated to be 0.32 eV, followed by a shallow barrier for the Se atom couple to the $C_{3h}$–Cu species (FS). The low rotation barrier of 0.32 eV elucidates the ease with which $C_s$–Cu species can rotate to $C_{3h}$–Cu species in the presence of Se on Cu(111) surface. The subsequent shallow barrier suggests that Se can readily bind with $C_{3h}$–Cu species to form the stable final state. The entire pathway is exothermic by 0.58 eV (FS). This explains the formation of the inner $C_{3h}$–Cu–$C_{3h}$ interlinked network as well as the peripheral $C_{3h}$ – Cu...Se interactions.

We have also calculated the energy barrier for breaking the interaction between $C_{3h}$–Cu and Se atom in the $C_{3h}$–Cu...Se terminal as shown in Supplementary Fig. 8. In view of kinetics, the $C_{3h}$–Cu terminals prefer to connect with Se atoms, in agreement with our experimental results (periphery $C_{3h}$–Cu...Se interactions around the network). But the low energy barrier (0.57 eV) still provides the Se atoms sufficient mobility so that bigger 2D OM networks can form at RT to 365 K. The reverse shallow barrier shows how easily the Se could couple to $C_{3h}$–Cu with an exothermic loss of 0.55 eV. The Arrhenius equation is defined as:

$$\nu = A \exp(-\Delta E/(k_B T)) \tag{1}$$

where the energy barrier $\Delta E$ is defined as the energy difference between the TS and IS, and $k_B$ is Boltzmann's constant. The prefactor $A$ is normally assigned the rule-of-thumb value of $10^{13}\,\mathrm{s}^{-1}$. To study the influence of the calculated energy barriers on the reaction kinetics, we define the Boltzmann ratio as:

$$r = \exp(-(\Delta E_1 - \Delta E_2)/(k_B T)) \tag{2}$$

where $\Delta E_1$ equals to 0.32 eV and $\Delta E_2$ equals to ≈0.58 eV as shown in Fig. 5b. As a result, the Boltzmann ratio $r \gg 1$ in quite a wide range of reaction temperature. Large values of $r$ indicates that the transformation from $C_s$ to $C_{3h}$ is preferred rather than the reverse transformation from $C_{3h}$ to $C_s$ after Se doping, which accounts for the high selectivity of $C_{3h}$ conformers after doping.

The decoupling of Se atom from the $C_s$–Cu species (possible process before the IS) could be excluded for the following reason: The overall energy of the $C_{3h}$–Cu network with peripheral $C_{3h}$–Cu...Se interactions was more stable compared with that of the $C_s$–Cu chain structure with terminal $C_s$–Cu...Se interactions. Single $C_{3h}$–Cu...Se or $C_s$–Cu...Se interaction is equal in energy for their intrinsic C–Cu...Se interaction. However, the number differs in different OM structures. Imaging there are equal numbers of $C_{3h}$ or $C_s$ precursors constituting the OM network or chain respectively, ideally, the $C_s$–Cu chain structure owns only two terminals available for further $C_s$–Cu...Se coupling while the $C_{3h}$–Cu network possesses a lot more peripheral terminals for further $C_{3h}$–Cu...Se coupling to stabilize the final structure. This quantitative difference explains that the overall energy for the $C_{3h}$–Cu network was more favorable compared with that for $C_s$–Cu chain structure when coupling to the Se atoms.

In experiments when we deposited the Se on the random OMs at low-temperature (≈200 K), the Se (or copper selenide) was also observed to locate beside the $C_s$–Cu species, in addition to form $C_s$–Cu...Se terminals (Supplementary Fig. 9). Therefore, we excluded the decoupling of Se atom from the $C_s$–Cu species for simplify, start with the configuration with Se beside the phenyl lobe, and try to rationalize our experimental observations with the DFT calculations by two steps: first, rotation of $C_s$–Cu species to $C_{3h}$–Cu species with the help of Se atoms locating beside the lobe and then followed by the coupling of Se atom attaching to the $C_{3h}$–Cu species to stabilize the final structure.

Extensive DFT calculations were performed on energy barriers for molecule migration on Cu(111) surface. In the migration of the $C_s$–Cu moiety, as illustrated in Supplementary Fig. 10, the moiety has the same orientation in the initial and the final states. Along one principal axis (indicated by red arrows) the migration was calculated to be 0.68 eV. While along another principal axis (indicated by black arrows), the sliding diffusion was calculated to be 0.29 eV. Migration along the third principal axis is, by symmetry, equivalent to the second (black) direction. Importantly, in both cases of the more stable initial and final states, the Cu adatoms were located in the more stable fcc sites, while in the transition states, the Cu adatoms were close to the less stable bridge sites. From a chemical point of view, the molecular moiety is nearly physisorbed on the surface (with distances between the triazine cores and the substrate of more than 3 Å). Thus, compared with the molecule, the chemisorbed Cu adatoms in the $C_s$–Cu moiety play a more important role in the energy diagram.

Similarly, in the migration as illustrated in Supplementary Fig. 11, the $C_{3h}$–Cu moiety also has the same orientation in the initial and the final states. Along the first principal axis (indicated by red arrows), the sliding diffusion barrier was calculated to be 0.48 eV, while along the second principal axis (indicated by black arrows) the sliding diffusion barrier was calculated to be 0.24 eV, respectively. Due to the alignment of the moiety, migration along the third principal axis is equivalent to the first, red, direction.

In addition to the migration barriers, the rotation barriers of $C_s$–Cu and $C_{3h}$–Cu moieties on Cu(111) were also calculated. Supplementary Fig. 12 illustrates the rotation process where the $C_s$–Cu or $C_{3h}$–Cu moiety rotates around the fixed triazine core. The rotation is important for two OM moieties to properly align before combination. For the $C_s$–Cu moiety, shown in the upper panel of Supplementary Fig. 12, the moiety in the final state is rotated clockwise by 120° around a normal to the surface going centered at triazine core, compared to the initial state. For the $C_{3h}$–Cu moiety, depicted in the lower panel of Supplementary Fig. 12, the final state is similarly rotated clockwise by 120° compared to the initial state. The energy barriers are calculated to be 0.25 eV and 0.30 eV for $C_s$–Cu and $C_{3h}$–Cu moieties, respectively, indicating how easily the $C_s$–Cu and $C_{3h}$–Cu moieties change their directions on the Cu(111) surface. Hence, neither the sliding diffusion nor the rotation diffusion processes are rate-limiting steps at RT.

As for the covalent interlinked products, further annealing of the C–Cu–C OM frameworks was also performed. Depicted in the Supplementary Fig. 13, demetallation of the OMs was observed after further annealing the sample at 470 K. The majority of the covalent products were 1D chains whether from the random (without Se doping) or ordered (with Se doping) OMs. After further annealing to 550 K, C–H activation was triggered, and porous N-doped graphene nanoribbons were formed through molecular folding. Similar porous graphene nanoribbons were also reported in the literature[19,26]. The corresponding STM images from random (without Se doping) or ordered (with Se doping) OMs followed by further annealing are shown in Supplementary Fig. 14. The reason is that at high temperatures (>470 K) on Cu(111), the energy barriers of transformation from either $C_{3h}$ to $C_s$ or $C_s$ to $C_{3h}$ could be overcome. What we finally got depends on the stability of the products. From the DFT calculations (Supplementary Fig. 15), the covalent 1D chain on Cu(111) is more stable than the 2D network. However, the focus of this study is 2D MOF formation at intermediate temperatures so we will not discuss this further.

In conclusion, a random phase consisting of $C_{3h}$ and $C_s$ conformers with both (L) and (R) handed chirality was obtained after deposition of the conformationally flexible precursor mTBPT on Cu(111) at RT to 365 K. Upon low coverage (0.01 ML) Se doping of mTBPT, we achieved the selectivity of the $C_{3h}$ conformer and improved the structural homogeneity, leading to the formation of an ordered 2D MOF. High selectivity of $C_{3h}$ conformer could be achieved whether pre- or post-exposing the sample to Se. This implies that an ordered 2D MOF can be fabricated regardless of the deposition sequence. Furthermore, the random phase consisting of both $C_{3h}$ and $C_s$ conformers

could be transformed to the ordered crystalline phase by low coverage (0.01 ML) Se doping at RT. Annealing to 365 K is not necessary for the transformation, but can accelerate the process and increase the proportion of hexagonal OM rings.

Through the combination of high-resolution STM/STS and nc-AFM at 4 K, we investigated the conformationally flexible precursor *m*TBPT on Cu(111) substrate and achieved the high topology selectivity of *m*TBPT by Se doping. NEB calculations in the absence or presence of Se have been employed to model the transformation between $C_s$–Cu and $C_{3h}$–Cu moieties on Cu(111) in order to explain the high topology selectivity of the $C_{3h}$ conformers by Se doping. These findings underscore the significance of Se doping in modulating the energy diagram of the moieties along the transformation path and achieving the desired topology selectivity in the system. The preferential transformation pathway in the presence of Se atom provides insights into the underlying mechanism of topology selectivity from both thermodynamic and kinetic point of view, which can prepare for the controlled synthesis of tailored metal-organic nanostructures.

## Methods

### STM/AFM characterization and sample preparation

The STM measurements were carried out in an ultrahigh vacuum (UHV) chamber with a base pressure of $2 \times 10^{-10}$ mbar. The whole system was equipped with an Omicron low-temperature STM interfaced to a Nanonis controller. A chemically etched tungsten tip was used for topographic measurements. The bias was applied to the sample, and the STM tip was grounded. STM images were analyzed with the WSxM software[47]. The Cu(111) substrate was prepared by several cycles of 1.5 keV Ar$^+$ sputtering followed by annealing to 680 K, resulting in clean and flat terraces separated by monatomic steps. After the system was thoroughly degassed, the *m*TBPT molecules (purchased from Adamas-beta, with purity >97%) were sublimated from a commercial Knudsen cell onto the Cu(111) substrate. The d$I$/d$V$ spectra were acquired by a lock-in amplifier while the sample bias was modulated by a 963 Hz, 20 mV (r.m.s.) sinusoidal signal under open-feedback conditions. The spectra are vertically shifted for clarity. The nc-AFM experiments were performed at 4 K in UHV. CO molecules for tip modification[45] were dosed onto the cold sample via a leak valve. The AFM is equipped with a qPlus sensor with a tungsten tip mounted on a quartz tuning fork (spring constant: 1800 Nm$^{-1}$, resonance frequency of $f_0 = 25.8$ kHz, $Q \approx 4.3 \times 10^4$, oscillation amplitude $\approx 40$ pm)[48]. The tip-height offsets $\Delta z$ for constant-height AFM images are defined as the offset in tip–sample distance relative to the STM set point at the Cu(111) surface. The negative values of $\Delta z$ correspond to the tip–sample distance decreased with respect to the STM set point. The *m*TBPT coverage of one monolayer refers to the most compact phase and equals 57 *m*TBPT molecules per 100 nm$^2$ in this system, and 1 ML Se is defined as 1 Se atom per surface Cu atom[32,49].

### DFT calculation

All the calculations were performed within the framework of DFT by using Vienna Ab Initio Simulation Package (VASP) code[50,51]. The method of projector augmented wave was used to describe the ion-core electron interactions[52,53]. The Perdew-Burke-Ernzerh of generalized gradient approximation (GGA) exchange-correlation functional was employed[54], and van der Waals corrections to the PBE density functional were also included using the DFT-D3 method of Grimme[55]. The plane wave was expanded to an energy cutoff of 420 eV for the models. The Cu(111) surface was modeled as a three-layer slab vertically separated by a vacuum region of 15 Å. For the transformation calculations, we used a $p(7 \times 7)$ surface unit cell in the absence of Se, whereas a $p(9 \times 9)$ surface unit cell was used in the presence of Se, together with a $5 \times 5 \times 1$ k-point grid in both cases. Reaction pathways were calculated with a combination of the climbing image nudge elastic band (CI-NEB)[56] and dimer[57] methods. The atomic structures

were geometrically relaxed until the forces on all unconstrained atoms were less than 0.03 eVÅ$^{-1}$. The IRI and IRI-π maps were calculated with B3LYP/6-31 G** (Gaussian 09)[58] combined with Multiwfn 3.8[46]. The DOS and PDOS maps of 2D OM network were calculated in vacuum with the Quantum Espresso software package[59] using the PBE exchange–correlation functional[54]. The AiiDAlab web platform[60] was used to perform and manage the calculations.

### Reporting summary

Further information on research design is available in the Nature Portfolio Reporting Summary linked to this article.

## Data availability

The data that support the findings of this study are available from the corresponding authors upon request. Source data are provided with this paper.

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

## Acknowledgements

The authors acknowledge financial support from the NRF CRP26 WBS A-8000421-03-00.

## Author contributions

L.C. and A. T.S.W. conceived the research. L.C. and T.G. performed the STM/AFM experiments and carried out the DFT calculations. All authors contributed to writing the manuscript.

## Competing interests
The authors declare no competing interests.
