## [Peer Review File · Nature Communications]

Topology Selectivity of a Conformationally Flexible Precursor through Selenium DopingREVIEWER COMMENTS

Reviewer #1 (Remarks to the Author):

In this manuscript under review, L. Cai et al demonstrate a Se-doping induced topology selection of a C-Cu-C organometallic frameworks from a molecular precursor mTBPT, which otherwise forms disordered structures because of coexistence of multiple adsorption conformers. By nc-AFM experiments and DFT calculations, the authors attribute the topology selectivity to the Se doping. Although this work excludes the discussion on possible selection of final covalent products, which is the common target of on-surface synthesis research, it may contribute to society of surface chemistry, molecular nanoscience and thus is suitable for publication on Nature Communications. Nevertheless, the discussion of the mechanism of topology selectivity raised some serious questions. I suggest its publication after the authors well addressing the comments listed below.

1. While the STM results clearly show the topology selection (2D mTBPT-Cu networks with chiral features), both nc-AFM and DFT calculation help less to understand the effect of Se-doping.

- In Fig. 3g and 3h, the AFM images show the existence of Se besides Cu; C-Cu...Se bonding was proposed (in line 188, "...which are speculated to be C-Cu...Se interactions"). This observation merely demonstrates that there exist Se adatoms and the adatoms "probably" interact to C-Cu species.
- Based on the above AFM observation, the author carried out DFT calculations on the adsorption energies of different molecular conformers (with or without Se coordination) and concluded that the conformation could be selected, considering the adsorption of Se at specific sites and with specific molecular adsorption. The selection of molecular conformers does not always give the topology selection of final 2D networks, because the latter depends simultaneously the linking nodes in between molecular monomers. The authors then proposed "...that the quickly diffusing Se atoms play an important role in regulating the precursor conformers and topology selectivity, ..." (line 158-159); this explanation is not convincing.

In short, neither AFM nor DFT give insights on the effect Se-doping.

2. The author used low coverage (<0.01 ML) Se doping, because high coverages led to the formation of copper selenide (in line 150-151).

- The authors need to define 1 ML;

- To give exactly how much the coverage is for the STM results with Se-doping.

- Related to Comment 1, I suggest the authors to investigate the mechanism of Se-doping by using different low coverages (<0.01 ML) or different ratios of mTBPT to Se.

3. The authors cited references 28 and 29. Both articles discussed Se doping or Se clusters in on-surface assembly (Ref. 29), but in my view were not so relevant to the topic talked in this manuscript. The author might need to search more literatures, which may help understand the Se adatoms/clusters effect occurring in this manuscript.

4. In line 93, 101 and 117, "hundreds of" / "more than 1000" molecules were included for statistical analysis. Please indicate exactly the counts in the corresponding histograms. Besides, I note that the error bars in Fig. 1f are quite similar; please double check the analysis result.

5. In line 95-96 "Different thermal treatment(s) including deposition of the". The author should present the experimental results in SI.

6. The text includes too many redundant phrases and typos, please revise it carefully. Some examples are listed below, to jus name a few.

"...at a wide temperature range from room temperature (RT) to ~365 K..." is repeated many times in line 49, 54, 120 and 240.

"These results could enrich the on-surface synthesis toolbox of conformationally flexible precursors, for the design of complex nanoarchitectures, and for future development of engineered nanomaterials and nanodevices." is exactly repeated in end of sections Abstract, Introduction and Conclusion. In line 28-29, "On-surface synthesis has been investigated in the past few years". On-surface synthesis has been developed over more than ten years; "...the resulting potential in applications electronics and magnetics." shall be "...the resulting potential applications in electronics and magnetics."

In line 78 "..., it is not difficult to find there are ..." should be "..., it is not difficult to find that there are

..."; in line 83 "(c) ... indicating the random phase are..." should be "(c) ... indicating that the random phase is..."

Reviewer #2 (Remarks to the Author):

Cai et al. investigated the influence of selenium in modulating the conformation of flexible motifs and obtained the ordered organometallic network with the same conformer on Cu(111). By utilizing STM/STS, nc-AFM and DFT calculations, the authors revealed the structure of products before and after Se doping, and determined the relative energy of the motifs at different adsorption sites. The complete dataset and control experiments provided insights into the effects of selenium doping on enhancing conformational selectivity. However, I am not convinced that this work offers the novelty and impact necessary to meet the standard of Nat. Commun. and raise the interest of a broad audience, due to the following reasons: 1) The influence of metal or ion doping on surface 2D network has been previously reported (J. Am. Chem. Soc. 2023, 145, 25, 13531–13536; ACS Nano 2023, 17, 11, 10938–10946); 2) Part of STM images exhibit poor quality, which does not support the authors' opinion; 3) The authors' discussion on the mechanism of conformation selectivity induced by Se doping is superficial and fails to meet the requirements of Nat. Commun. My suggestion is to publish it in a more specialized journal. I list below some points that the authors should address:

- 1) It is evident that Fig. 1a-c display multiple tips, and the resolution of these images is insufficient to provide strong support for the authors' statements. Additionally, Fig. 1d-e demonstrate prominent noise levels.
- 2) In 118 on page 7. Do these uncertain structures contain Cs conformations? Is the formation of C3h conformation more favorable within the island?
- 3) From Figure S1b, it appears that the proportion of non-hexagon monomers in the 2D networks has increased at RT. The authors should address whether thermal deposition has the potential to increase the proportion of hexagonal organometallic rings.
- 4) What are the brighter and larger dots at the edge of the network in Fig. 2d, as opposed to the darker dots (Br adatoms)?
- 5) In 146 on page 8. At higher temperature, molecules have higher mobility on the surface, which is more conducive to the self-healing of 2D OM networks, and achieve a more thermodynamically stable structure. Why is it inconsistent with the conclusion of this work?
- 6) In 154 on page 8. Is the diffusion barrier of copper selenide the primary factor hindering the formation of a large-scale network, or is it the insufficient coverage of the precursors?
- 7) In the penultimate paragraph on page 8, the authors pointed that due to the limited diffusion energy barrier of copper selenide, it is impossible to generate a large two-dimensional network. However, the last paragraph on page 8 noted that Se atoms have a low migration energy barrier on Cu(111) and can spread rapidly, thus regulating product selectivity. This is contradictory, and the author should be consistent.
- 8) Why does the dI/dV spectrum located on the molecule have a peak at about +1.0 V (red curve).
- 9) In 215 on page 12. The data "0.12 eV" here is incorrect, the value should be 0.08 eV.
- 10) The author needs to provide further characterization to more directly illustrate the existence of C-Cu...Se intermediates.
- 11) Regarding the DFT calculations of the mechanism on topology selectivity, it would be one-sided and meaningless to solely analyze the energy advantage of the monomer. The interaction between molecules and substrate, and the migration of structures on the surface, etc., play crucial roles in determining the formation of the 2D networks. This is particularly true for Cu(111) substrate, where the interaction is notably strong, and the energy barrier for molecule migration on the surface cannot be overlooked.
- 12) In the last paragraph on page 12. The analysis provided here lacks justification, because the formation of periodic structures is not a prerequisite for 2D OM networks on surface. Previous research has demonstrated that similar energy of the conformations of a precursor can cause a loss of coupling selectivity in 2D structures, leading to disorder (Chem. Commun. 2010, 46, 7157-7159; Nat. Mater.

Reviewer #3 (Remarks to the Author):

In this manuscript, the authors used Se-doping strategy to achieve the topology selectivity of a conformationally flexible precursor and construct an ordered 2D metal organic network on Cu(111) surface. The main conclusion is supported by high-resolution STM and nc-AFM investigations. Furthermore, a plausible mechanism is proposed based on the experimental examinations as well as DFT calculation.

The findings introduce an important approach to the growing field of on-surface chemistry, which opens new avenues towards the synthesis of chiral low-dimensional materials. The manuscript is overall well written, is based on high-quality experimental data, an insightful analysis and touches a timely topic.

Prior to publication, a series of points deserves further attention:

1. The annealing temperatures should be presented in the corresponding STM images (Figures 1 and 2).
2. The chemical structure of organometallic network can be fitted in Figure 2 (partially fitted). That may be reader-friendly.
3. The STS measurements of network are strongly affected by the surface state of Cu(111). It is helpful to show the gas-phase DOS of this network to have a better understanding of its electronic properties.
4. The DFT calculations partially explained the transformation of random phase to ordered network. The topology selectivity of conformations is the key point of this research, this crucial issue requires a more rigorous treatment. The transformation from Cs to C3h conformer by flipping one of m-bromophenyl groups was confirmed by author's experimental data. What is the flipping process, and the role of Se atom? The Se atoms improve the reversibility of C-Cu-C bond, just like the role of I, Br, and Cl atoms during the on-surface synthesis?

Reviewer #4 (Remarks to the Author):

See attached.

Referee report on the manuscript “Topology Selectivity of a Conformationally Flexible Precursor by Selenium Doping” submitted by Cai et al. for publication in Nature Communications.

In this manuscript (MS), Cai et al. present an intriguing way of controlling the conformationally of organometallic (OM) structures made of triphenylazine (mTBPT) precursor on Cu(111) using selenium. A combination of STM, nc-AFM and DFT is used to identify the chemistry of the OM structure, establish a statistical analysis of the different conformers and rationalize the effect of selenium. After deposition of mTBPT on Cu(111), two different phases and its chiral fashions are found: R-Cs, L-Cs, R-C3h and L-C3h. The statistical analysis, as expected, shows that there is no selectivity for one of these phases and their almost equally present. Upon introduction of Se (independently of the deposition sequence), the authors claim that the presence of R/L-Cs conformers is significantly reduced and only R/L-C3h is found. This transition and its perspective are very interesting. Tuning the symmetry of a system and selecting a specific among various options is of great interest. The explanation of this effect and the identification of the underpinning mechanism would enable the translation of this work on to different systems.

However, the experimental demonstration and reasons why the presence of an amount of Se atoms on Cu(111) can produce this transition are unclear. The four phases expected from the deposition of mTBPT result from its conformational flexibility and its characteristic prochiral properties that mainly play during the deposition process. To have the transition, all the molecules in a Cs form would have to transform into the C3h form by reversing one of the m-bromophenyl groups, as rightly reported on page 12. This transition seems very unlikely from an energetical point of view due to the barrier to overcome; the cited references 10 (not relevant),44 (refers to the gas phase),45 (not relevant) provide no clear explanation.

The current provided DFT simulations rely on the assumption that the symmetry of the two-dimensional network can be ducted from the orientation of single molecule calculations and their positioning with respect to the bridging Cu atoms. The calculation suggest that C3h symmetry is the more stable than Cs by 0.08 eV in the absence of Se. This trend is the same with the consideration of Se (0.14 eV). These calculations suggest that both assemblies should be present on the surface and they do not show the selectivity. I would expect a much larger enthalpic loss for Cs and I suspect that it can be revealed from periodic 2D network simulations. In addition, the experiments suggest that the conformer information of the molecule changes when adding Se as the Cs conformers vanish and have to transform to C3h.

Therefore, I would only recommend accepting this MS after thoroughly reviewing the theoretical elaboration and after obtaining further experimental results showing, on the same surface area, the transformation of the Cs form into the C3h form due to the close presence of Se atoms.

In the following additional remarks and comments.

General remarks:

The quality of the Figures is lacking. In Figure 1f, the structural model is very pixilated. However, understanding the different conformers is key to appreciate the work. Moreover, Figure 4 should be redesigned; there is too much information in the Figure.

All experiments are conducted well, but some noise and scratches are present on the images (e.g. Figure 1d) that could be cleaned up.

I suggest deleting the term “wide” relating to the temperature used (room temperature to 365 K). The amount of Se atoms must be indicated carefully: “low coverage” is not clear enough. I suggest modifying Scheme 1 to clearly include the four species, as Scheme 1 of Ref. 45.

Comments:

On the transition of the conformer:

Figure 1 demonstrates that all conformers and their chiral versions (R/L-C3h and R/L-Cs) exist almost equally. The addition of Se to this configuration transforms the monomers to change to C3h exclusively (Figure 2). This means that either the phenyl rotates with respect to the triazine core or that the radical migrates along the C-C backbone. The authors have to investigate the catalytic effect of Se. Nudged elastic band calculations in the absence or presence of Se can be employed to model this rotation. If the activation barrier drastically decreases, it could be the key to understand the experimental results.

DFT calculations in Figure 2 of the adsorbed 2D OM phase and the unit cell:

The authors write that experimentally the lattice parameters are: $a = b = 23.2 \text{ \AA}$. However, the DFT simulated unit cell has the lattice parameters: $a = b = 22.28 \text{ \AA}$. While I do not believe that the change of surface orientation between DFT and experiment is too important, the lattice strain is 4.13 %, which is quite large. This strong compressive strain on the network will likely make the periodic system wavy to release the strain. The authors use this wavy nature to explain the different contrasts between the monomers in the unit cell. However, it seems to me more like a DFT artifact due to the strong compression. Can the authors improve the simulation and reduce the strain on OM structure? Could the authors provide the lattice parameters of the network in gas-phase and compare them to the current simulations?

On the alternating contrast difference:

In Figure 3, the authors show highly resolved images of the OM structure using nc-AFM. The dimmer molecule of the unit cell seems very flat. In contrast the triazine core of the other molecule appears bright. The same is observed in the STM images. Beyond a possible wavy nature of the OM network, which comes at an enthalpic cost and should make it less stable, are there other possibilities to recreate this contrast? Could the triazine core act as a Se trap and the Se intercalates below as a single atom?

DFT simulation of intact molecules Page 11 line 202-210:

The authors have carried out simulations of the two different conformers before dehalogenation in the absence and presence of Se. “These results suggest that the presence of Se atoms can effectively modulate the energy landscape of the system, favoring the formation of specific conformers”. While the statement is true, I do not see how this relates to the work and how it improves the rationalization of the mechanism. The precursors immediately dehalogenate upon adsorption. Therefore, the experiment will not resemble the simulations. Moreover, the adsorption site of the Se atom in Figure S5d is quite odd. There should be some repulsion between the Se and the halogens, which could provide an enthalpic loss and could be the reason for the 0.11 eV difference.

DFT simulation of dehalogenated molecules with ad atoms in different adsorption sites without Se atoms (Figure 4):

The authors have carried out DFT calculations to compute the energy difference between different adsorption geometries. “By careful comparison between the calculated 2D MOF in Figure 2g-h and the C3h-Cu moieties in Figure 4a-f, it was observed that the network is composed of C3h-I and C3h-II moieties, as highlighted by the red and black rectangle” Could the authors visualize this? Moreover,

the argumentation “that the network is composed of C3h-I and C3h-II moieties” relies on the heavily strained system, which makes it questionable. Can the authors please comment how they extract relevant information from the single molecule adsorption geometries for the 2D OM network? The calculation as of now suggest that almost all configurations are valid as their energies are quite comparable <0.2 eV.

DFT simulation of dehalogenated molecules with ad atoms in different adsorption sites with Se atoms (Figure 4):

The authors compared the aforementioned trend of energies without Se atoms to similar simulations with Se atoms. They state “This regulation resulted in the favouring of specific conformers (C3h-I and C3h-II in Figure 4m and 4n), exhibiting lower energies compared to other configurations and indicating increased stability, thus finally facilitated the formation of the desired 2D MOFs with high topology selectivity.” The question is how would the structure in Figure 4m,n form the 2D OM network, if the Se atoms are placed in such a way that incoming monomers cannot connect to the Cu atom because the Se atom blocks it. Most of the proposed structures are having this issue.

I believe the authors should focus on explaining how the transformation from Cs to exclusively C3h occurs. Se atoms seem to catalyse this transition and it should be reflected in a kinetic point of view through the activation barriers of the dihedral rotation. The presented picture on the thermodynamic site seems to evaluate the effect of the Se on the adsorption of the monomer on Cu(111) (some structures are less planar than others, e.g. Figure 4n compared to Figure 4u,p) and not the formation of the 2D network. I could imagine that the Se decouples the organic backbone from the surface and prevents hybridization, which would facilitate rotation as the molecule-substrate interactions are weaker.

Computational methods section 4):

The authors do not state the construction of the surface slaps. Moreover, the $2 \times 2 \times 1$ k-point grid seems quite low to make comparison in the 100 meV range as the total energy values may still fluctuate with increasing k-points.

Reviewer #5 (Remarks to the Author):

I co-reviewed this manuscript with one of the reviewers who provided the listed reports as part of the Nature Communications initiative to facilitate training in peer review and appropriate recognition for co-reviewers.

1) Reviewer #1 (Remarks to the Author):

In this manuscript under review, L. Cai et al demonstrate a Se-doping induced topology selection of a C-Cu-C organometallic frameworks from a molecular precursor mTBPT, which otherwise forms disordered structures because of coexistence of multiple adsorption conformers. By nc-AFM experiments and DFT calculations, the authors attribute the topology selectivity to the Se doping. Although this work excludes the discussion on possible selection of final covalent products, which is the common target of on-surface synthesis research, it may contribute to society of surface chemistry, molecular nanoscience and thus is suitable for publication on Nature Communications. Nevertheless, the discussion of the mechanism of topology selectivity raised some serious questions. I suggest its publication after the authors well addressing the comments listed below.

Reply: We thank the reviewer for the positive comments. Regarding the reviewer's comment on final covalent products, further annealing of the C-Cu-C 2D MOF was performed and final covalent products were also studied. As the scheme S1 shown below, demetallation of the organometallics was observed after further annealing the sample at 470 K. The majority of the fabricated covalent products were 1D chains no matter from the random (without Se doping) or ordered (with Se doping) organometallics.

Further annealing the sample to 550 K, C-H activation was triggered, and the porous N-doped graphene nanoribbon was formed through molecular folding. A similar porous graphene nanoribbon was also reported in the literature [J. Am. Chem. Soc. 2017, 139, 37, 12976–12984; ACS Nano 2021, 15, 4617–4626].

The corresponding STM images from random (without Se doping) or ordered (with Se doping) organometallics followed by further annealing are also shown below in Figure S11. At high temperatures (> 470 K) on Cu(111), the energy barriers of transformation from whether C_{3h} to C_s or C_s to C_{3h} could be overcome. What we finally got depends on the stability of the products. From the DFT calculations (Figure S12), the covalent 1D chain on Cu(111) is more stable than the 2D network. Since it is out of the scope of our main point *Topology Selectivity by Selenium Doping*, we simply discussed the covalent structures in the revised manuscript and put the following figures in the Supporting Information.

Scheme summarizes the thermal reaction pathways described.

Scheme S1. The blue dashed rectangle framed the revised Scheme 1 in the manuscript. No matter random (without Se doping) or ordered (with Se doping) organometallics went through demetallation to form 1D covalent interlinked chains (470 K), and finally C–H activation was triggered to form porous N-doped nanoribbon with further annealing (550 K).

Figure S11. STM images after further annealing from random (without Se doping) or ordered (with Se doping) organometallics. (a) Large-scale and (b) close-up STM images showing 1D covalent interlinked chains (majority) after annealing at 470 K from the random organometallics (without Se doing). (c) Large-scale and (d) close-up STM images showing the N-doped porous graphene nanoribbon after further annealing at 550 K (without Se doing). (e) Large-scale and (f) close-up STM images showing 1D covalent interlinked chains (majority) after annealing at 470 K from the ordered 2D organometallics (with Se doing). (g) Large-scale and (h) close-up STM images showing the N-doped porous graphene nanoribbon after further annealing at 550 K (with Se doing). STM parameters: constant current, (a-c, g) $U = -1$ V, $I = 100$ pA; (d) $U = -500$ mV, $I = 100$ pA; (e) $U = 1$ V, $I = 100$ pA; (f) $U = 500$ mV, $I = 50$ pA; (d) $U = -500$ mV, $I = 50$ pA.

Figure S12. DFT calculations on the covalent interlinked 1D chain and 2D network. DFT relaxed models of covalent interlinked (a) 1D chain and (b) 2D network in the gas phase. DFT relaxed models of covalent interlinked (c) 1D chain and (d) 2D network on Cu(111). The

calculated energies were also indicated in units of eV. Brown, white, gray, and blue balls represent C, H, N, and Cu substrate, respectively.

1. While the STM results clearly show the topology selection (2D mTBPT-Cu networks with chiral features), both nc-AFM and DFT calculation help less to understand the effect of Se-doping.

i. In Fig. 3g and 3h, the AFM images show the existence of Se besides Cu; C-Cu...Se bonding was proposed (in line 188, "...which are speculated to be C-Cu...Se interactions"). This observation merely demonstrates that there exist Se adatoms and the adatoms "probably" interact to C-Cu species.

Reply: We thank the reviewer for pointing out this issue. The interaction region indicator (IRI) analysis on the C-Cu...Se moiety was performed to investigate the intermolecular interactions. To illustrate the nature of interactions between Se adatoms and C-Cu species, we used a finite C-Cu moiety with one Se atom terminated as a model system in the IRI analysis. IRI is a new real-space function that can clearly reveal both chemical bonds and weak interactions in chemical systems [Chem. Methods 2021, 1, 231–239]. As illustrated in the IRI isosurface below shown in Figure S7, both covalent bond and weak interaction regions are nicely revealed by the blue and green isosurfaces, respectively, according to the standard coloring method. It thus demonstrates that there is nonnegligible interaction between the Se adatom and the C-Cu species.

Figure S7. The IRI map of finite C-Cu...Se moiety. Upper: The IRI isosurface map of finite C-Cu...Se moiety showing the interactions between the Se adatom and the C-Cu species. Blue and green isosurfaces represent the covalent interactions and weak interactions, respectively. Lower: Standard coloring method. Brown, white, gray, pink, and yellow balls represent C, H, N, Cu, and Se atoms, respectively. The scales of color bars are given in a.u.

ii. Based on the above AFM observation, the author carried out DFT calculations on the adsorption energies of different molecular conformers (with or without Se coordination) and concluded that the conformation could be selected, considering the adsorption of Se at specific sites and with specific molecular adsorption. The selection of molecular conformers does not always give the topology selection of final 2D networks, because the latter depends simultaneously the linking nodes in between molecular monomers. The authors then proposed “...that the quickly diffusing Se atoms play an important role in regulating the precursor conformers and topology selectivity, ...” (line 158-159); this explanation is not convincing. In short, neither AFM nor DFT give insights on the effect Se-doping.

Reply: Before we discuss the DFT calculations, we shall highlight the nc-AFM data. The nc-AFM of the 2D *m*TBPT–Cu networks shows more details especially the clear skeleton of the framework, displaying the molecular buckling and height difference (Figure 3). More importantly, the nc-AFM of the network periphery (Figure 3g) offers valuable clues that can shed light on the underlying mechanism of regulating crystallization through Se doping. As illustrated in Figure S6 (also shown below), terminal interactions to the periphery molecules were indicated by white ellipses. Considering that there is no such interaction at the periphery molecules before Se doping (surface stabilized radicals as shown in Figure S2b), we attribute the interaction to Se. Combined with the high-resolution STM and nc-AFM images shown below, we assign them (depicted by white ellipses) to C–Cu...Se interactions as highlighted in Figure S6e.

Figure 3. The structure of 2D MOFs on Cu(111). (a) The nc-AFM frequency shift image and (b) current image acquired simultaneously of the “R” domain. (c) The nc-AFM frequency shift image and (d) current image acquired simultaneously of the “L” domain. (e) High-resolution of nc-AFM frequency shift image and (f) current image acquired simultaneously indicating the C–Cu–C interlinking. (g) The nc-AFM frequency shift image showing the periphery interactions. (h) The corresponding current image acquired at the same position of panel (g). The C–Cu...Se interactions are indicated by white ellipses, and detached Br atoms by red ellipses, respectively. Scanning parameters: (a-g) constant height mode using a CO-terminated

tip at a bias voltage of 0 V; (h) constant current mode with $U = 1$ V, $I = 100$ pA. Images (a-g) were recorded at different tip offsets Δz (a, b, $\Delta z = -250$ pm; c, d, $\Delta z = -240$ pm; e, f, g, $\Delta z = -260$ pm) with respect to an STM set point ($U = 1$ V, $I = 100$ pA).

Figure S6. Periphery interactions C–Cu...Se of the network. (a) The current image and (b) corresponding nc-AFM frequency shift image of R-network acquired at the same position show the periphery interactions. (c) The current image and (d) corresponding nc-AFM frequency shift image of L-network acquired at the same position show the periphery interactions. White ellipses indicate the terminal interactions to the periphery molecules. (e) High-resolution of nc-AFM frequency shift image superimposed with the model indicates the C–Cu...Se interaction of the network periphery. Scanning parameters: (a, c) constant current mode with $U = 1$ V, $I = 100$ pA. (b, d, e) constant height mode using a CO-terminated tip at a bias voltage of 0 V, images were recorded at the tip offset $\Delta z = -250$ pm with respect to an STM set point ($U = 1$ V, $I = 100$ pA).

Figure S2b. (b) The close-up STM image shows the random phase formed by *mTBPT* on Cu(111) at RT. STM parameters: constant current, (b) $U = 500$ mV, $I = 100$ pA.

For the theoretical part, we performed additional DFT calculations to unravel the underlying mechanism, especially the energy barriers of C_s transformed to C_{3h} with and without Se. As shown in the revised Figure 4a (also shown below), the energy barrier of C_s transformed to C_{3h} without Se was calculated to be 0.52 eV, which can be overcome at RT. Considering the similar energies of C_s and C_{3h} conformers, it is expected that random organometallics were formed with no obvious selectivity.

However, with Se doping, the transformation from C_s to C_{3h} is a two-barrier process, where configuration with Se standing beside the phenyl lobe constitutes a shallow intermediate state and the configurations with Se connecting to the C_s -Cu and C_{3h} -Cu constitute initial and final states. The energy barrier of C_s transformed to C_{3h} with Se was calculated to be 0.60 eV from the initial state to the intermediate state, and 0.48 eV for the second barrier from the intermediate state to the final state (Figure 4b). Whereas the converse energy barrier of C_{3h} transformed to C_s with Se was calculated to be 0.99 eV with no energetically favorable intermediate state. The Arrhenius equation is defined as:

$$v = A \exp(-\Delta E/k_B T),$$

where the energy barrier ΔE is defined as the energy difference between the TS and IS, and k_B is Boltzmann's constant. The prefactor A is assigned the rule-of-thumb value of 10^{13} s^{-1} . To study the influence of the calculated energy barriers on the reaction kinetics, we define the Boltzmann ratio as:

$$r = \exp(-(\Delta E_1 - \Delta E_2)/k_B T)$$

where ΔE_1 equals to 0.60 eV and ΔE_2 equals to 0.99 eV as shown in Figure 4. As a result, the Boltzmann ratio $r \gg 1$ in quite a wide range of reaction temperature. Large values of r indicates that the transformation from C_s to C_{3h} is preferred rather than the reverse transformation from C_{3h} to C_s after Se doping, which explains the high selectivity of C_{3h} conformers after doping.

We have nevertheless softened the conclusion by the inclusion of this sentence:

"...based on our observations and calculations, it is likely that the Se atoms play an important role in regulating the precursor conformers and topology selectivity."

Figure 4. Energy diagrams for the transformation from C_s to C_{3h} (a) without and (b) with Se doping. The structural models are given for the initial, transition, intermediate and final states, respectively. The energy scale is not linear. Energies are given in units of eV. Brown, white, gray, green, blue, and pink balls represent C, H, N, Se, Cu substrate, and Cu adatoms, respectively.

2. The author used low coverage (<0.01 ML) Se doping, because high coverages led to the formation of copper selenide (in line 150-151).

i. The authors need to define 1 ML;

Reply: 1 ML Se is defined as 1 Se atom per surface Cu atom, which is in accordance with the previous study [ChemPhysChem 2016, 17, 2137–2145]. We also revised this in the Methods and Materials section.

ii. To give exactly how much the coverage is for the STM results with Se-doping.

Reply: We revised the low coverage Se doping to more exact parameters of 0.01 ML in the revised manuscript based on counting 10 images.

iii. Related to Comment 1, I suggest the authors to investigate the mechanism of Se-doping by using different low coverages (<0.01 ML) or different ratios of mTBPT to Se.

Reply: According to the reviewer's suggestion, we performed additional experiments with different coverage of Se and different ratios of *m*TBPT to Se. All of these samples were thermally treated at 365 K for 2 hours after co-deposition of *m*TBPT and Se.

When 0.005 ML Se and 0.28 ML *m*TBPT were mixed on Cu(111), there coexisted C_s and C_{3h} conformers, forming organometallic chains and networks as shown in Figure S3a and S3b (also shown below). After increasing the coverage of Se to 0.01 ML (Figure S3c and S3d), only the C_{3h} conformer was observed, leaving some small copper selenide domains beside. That is to say, the amount of Se is related to the stabilization of the C_{3h} conformer. Therefore, we hypothesize that the C_{3h} conformer could be stabilized with the doped Se atoms. Considering the low diffusion barrier of Se atoms on Cu(111) (0.107 eV) [ChemPhysChem 2016, 17 (14), 2137–2145], it is plausible that the Se atoms play an important role in regulating the precursor conformers and topology selectivity. More details about how the Se atoms influence the energy diagram are provided in the DFT discussion.

In order to further demonstrate this hypothesis, the coverage of *m*TBPT was increased to 0.70 ML (Figure S3e and S3f), the network grew bigger, and the chain structure composed of the C_s conformer was observed again for the insufficient Se. With increasing Se to 0.015 ML (Figure S3g and S3h), the C_s conformer was observed less.

Figure S3. STM images with different ratios of Se to *m*TBPT after thermal treatment (365 K for 2 h). (a) Large-scale and (b) close-up STM images showing coexistence of 1D chains and 2D networks after co-deposition of 0.005 ML Se and 0.28 ML *m*TBPT. (c) Large-scale and (d) close-up STM images showing 2D networks and small copper selenide domains after increasing the coverage of Se (0.01 ML Se and 0.28 ML *m*TBPT). (e) Large-scale and (f) close-up STM images showing 2D networks (growing larger) and some 1D chains forming after further increasing the coverage of *m*TBPT (0.01 ML Se and 0.70 ML *m*TBPT). (g) Large-scale and (h) close-up STM images showing 2D networks with negligible 1D chains after adding

more Se (0.015 ML Se and 0.70 ML *m*TBPT). STM parameters: constant current, (a, g, h) $U = 1$ V, $I = 50$ pA; (b) $U = 200$ mV, $I = 100$ pA; (c, e, f) $U = 500$ mV, $I = 50$ pA; (d) $U = 400$ mV, $I = 100$ pA.

3. The authors cited references 28 and 29. Both articles discussed Se doping or Se clusters in on-surface assembly (Ref. 29), but in my view were not so relevant to the topic talked in this manuscript. The author might need to search more literatures, which may help understand the Se adatoms/clusters effect occurring in this manuscript.

Reply: After searching more literature, we have added two relevant references.

The influence of metal or ion doping on surface 2D network:

Ref. (1) Assembling Surface Molecular Sierpiński Triangle Fractals via K^+ -Invoked Electrostatic Interaction [J. Am. Chem. Soc. 2023, 145, 25, 13531–13536].

Ref. (2) Tuning Chirality of Self-Assembled PTCDA Molecules on a Au(111) Surface by Na Coordination [ACS Nano 2023, 17, 11, 10938–10946].

These references could to some extent help us understand the doping effects, but specifically Se adatoms/clusters effect on surface 2D network has not yet been studied to our knowledge, and this topic is important in view of the increased interest in selenides.

4. In line 93, 101 and 117, “hundreds of” / “more than 1000” molecules were included for statistical analysis. Please indicate exactly the counts in the corresponding histograms. Besides, I note that the error bars in Fig. 1f are quite similar; please double check the analysis result.

Reply: We revised the histograms to include the exact counts in the corresponding histograms. In line 93: There is no obvious selectivity of *m*TBPT conformers from a statistical analysis of 1200 molecules as shown in Figure 1g, which explains the formation of the random C–Cu–C organometallic nanostructure.

Line 101: In Figure 2(c), statistical analysis of 1200 molecules showing high selectivity of *m*TBPT forming C_{3h} moieties on Cu(111) after selenium doping.

Line 117: From the statistical analysis of 1200 molecules in Figure 2c, the selectivity of C_{3h} conformers is quite high (close to 90%).

We also revised the error bars in Figure 1g (the original Figure 1f) as shown below.

Figure 1g. Statistical analysis of 1200 molecules from 20 images showing no obvious selectivity of *m*TBPT conformers on Cu(111).

5. In line 95-96 “Different thermal treatment(s) including deposition of the”. The author should present the experimental results in SI.

Reply: According to the reviewer’s suggestion, the experimental results were presented in SI: Different thermal treatments including deposition of the precursor at RT or higher temperature (365 K) did not show any selectivity towards a specific conformer (Figure S1).

Figure S1. STM images showing the random phase formed after deposition of *m*TBPT on Cu(111) at (a) RT and (b) 365 K. STM parameters: constant current, (a) $U = -1$ V, $I = 100$ pA; (b) $U = 1$ V, $I = 100$ pA.

6. The text includes too many redundant phrases and typos, please revise it carefully. Some examples are listed below, to jus name a few.

“...at a wide temperature range from room temperature (RT) to ~365 K...” is repeated many times in line 49, 54, 120 and 240.

“These results could enrich the on-surface synthesis toolbox of conformationally flexible precursors, for the design of complex nanoarchitectures, and for future development of engineered nanomaterials and nanodevices.” is exactly repeated in end of sections Abstract, Introduction and Conclusion.

In line 28-29, “On-surface synthesis has been investigated in the past few years”. On-surface synthesis has been developed over more than ten years; “...the resulting potential in applications electronics and magnetics.” shall be “...the resulting potential applications in electronics and magnetics.”

In line 78 “..., it is not difficult to find there are ...” should be “..., it is not difficult to find that there are ...”; in line 83 “(c) ... indicating the random phase are...” should be “(c) ... indicating that the random phase is...”

Reply: We have revised the listed phrases to remove redundancies.

e.g. “...at a wide temperature range from room temperature (RT) to ~365 K...” is revised to “at RT to 365 K”.

In the end of Introduction and Conclusion, the sentences are also rewritten.

Introduction: “These results have the potential to enrich the on-surface synthesis toolbox of conformationally flexible precursors, encouraging the design of intricate nanoarchitectures.”

Conclusion: “These results could give us new methods of on-surface synthesis involving conformationally flexible precursors, which could inspire the design of intricate nanoarchitectures/nanomaterials, and may contribute to the development of engineered nanomaterials/nanodevices.”

In line 28-29, the sentence is revised to “On-surface synthesis has been investigated in the past decades owing to the synthetic accessibility of novel nanostructures.”

In line 78 and 83, for Figure 1a-c, we have performed further STM experiments and replaced the original Figure 1a-c (random organometallics composed of partially debrominated *m*TBPT precursors) with the revised Figure 1a-d (intact precursors after deposition at lower temperature of 90 K) of high quality. Thus, we have rewritten this paragraph and the figure caption.

We also carefully looked through the whole manuscript thoroughly, and once again thank the reviewer for the careful reading of the manuscript.

Reviewer #2 (Remarks to the Author):

Cai et al. investigated the influence of selenium in modulating the conformation of flexible motifs and obtained the ordered organometallic network with the same conformer on Cu(111). By utilizing STM/STS, nc-AFM and DFT calculations, the authors revealed the structure of products before and after Se doping, and determined the relative energy of the motifs at different adsorption sites. The complete dataset and control experiments provided insights into the effects of selenium doping on enhancing conformational selectivity. However, I am not convinced that this work offers the novelty and impact necessary to meet the standard of Nat. Commun. and raise the interest of a broad audience, due to the following reasons: 1) The influence of metal or ion doping on surface 2D network has been previously reported (J. Am. Chem. Soc. 2023, 145, 25, 13531–13536; ACS Nano 2023, 17, 11, 10938–10946); 2) Part of STM images exhibit poor quality, which does not support the authors' opinion; 3) The authors' discussion on the mechanism of conformation selectivity induced by Se doping is superficial and fails to meet the requirements of Nat. Commun. My suggestion is to publish it in a more specialized journal. I list below some points that the authors should address:

Reply: For main point 1) *The influence of metal or ion doping on surface 2D network has been previously reported (J. Am. Chem. Soc. 2023, 145, 25, 13531–13536; ACS Nano 2023, 17, 11, 10938–10946):*

In the first reference [J. Am. Chem. Soc. 2023, 145, 25, 13531–13536] entitled “Assembling Surface Molecular Sierpiński Triangle Fractals via K⁺-Invoked Electrostatic Interaction”, the authors transformed C–H···Cl–C hydrogen bonds to electrostatic interaction between the K cation and electronically polarized Cl atoms, thus achieving the Sierpiński triangle fractal assembly.

In the second reference [ACS Nano 2023, 17, 11, 10938–10946] entitled “Tuning Chirality of Self-Assembled PTCDA Molecules on a Au(111) Surface by Na Coordination”, the authors transformed hydrogen bond to coordination bond, thus achieving the chirality tuning.

Our work is distinct from the above references. In our study, the interaction is about C–Cu–C organometallics. The high selectivity of conformation to form 2D MOFs is achieved by the transformation of the precursor skeleton from the C_s to the C_{3h} conformer, and the underlying mechanism is that the doped Se atoms regulate the transformation energy barriers.

For main point 2) *Part of STM images exhibit poor quality, which does not support the authors' opinion:*

We have performed further STM experiments and revised the Figure 1 in the revised manuscript with new STM images. All the STM images in the revised Figure 1 has been replaced with high quality ones. The original Figure 1a-c indicating random organometallics composed of partially debrominated *m*TBPT precursors were replace by revised Figure 1a-d depicting the intact precursors after deposition at lower temperature of 90 K. Original Figure 1d and 1e indicating random organometallics composed with fully debrominated *m*TBPT precursors were replaced by revised Figure 1e and 1f with high quality (also shown below).

Figure 1. Intact conformers and the random phase formed by *mTBPT* on Cu(111). (a) Overview and (b) close-up STM images showing isolated monomers and self-assembled domains with limited order after deposition of *mTBPT* on Cu(111) at 90 K. The close-packed directions of Cu(111) are marked by white arrows. (c, d) High-resolution STM images indicating the intact conformers R- C_s -, L- C_s -, R- C_{3h} -, and L- C_{3h} -*mTBPT* together with their schematic models. (e) Overview and (f) close-up STM images after annealing to 365 K, showing the random phase of C–Cu–C organometallics formed by fully debrominated *mTBPT* precursors and Cu adatoms. (g) Statistical analysis of 1200 molecules indicating no obvious selectivity of *mTBPT* conformers on Cu(111). STM parameters: constant current, (a, e, f) $U = 1$ V, $I = 100$ pA; (b-d) $U = -1$ V, $I = 100$ pA.

For main point 3, we performed additional experiments with different coverage of Se and different ratios of *mTBPT* to Se to investigate the mechanism of Se doping, as discussed in the reply to Reviewer 1 (see Figure S3). The discussion on the mechanism of conformation selectivity induced by Se doping is now more detailed.

In addition to the detailed discussion on Se doping and precursor coverage on topology selectivity, we also performed additional DFT calculations of the energy barriers of C_s transformed to C_{3h} with and without Se, as described in the response to Reviewer 1 (Figure 4). We hope the manuscript now meets the requirements of *Nature Communications*.

Additional points:

1) It is evident that Fig. 1a-c display multiple tips, and the resolution of these images is insufficient to provide strong support for the authors' statements. Additionally, Fig. 1d-e demonstrate prominent noise levels.

Reply: We have performed further STM experiments and replaced all STM images in the revised Figure 1 (shown above). The noise level in Figure 1d-e is now significantly reduced.

2) In 118 on page 7. Do these uncertain structures contain C_s conformations? Is the formation of C_{3h} conformation more favorable within the island?

Reply: There are probably few C_s conformations in the uncertain structures. We have no direct evidence that the C_{3h} conformation is more favorable within the island.

3) From Figure S1b, it appears that the proportion of non-hexagon monomers in the 2D networks has increased at RT. The authors should address whether thermal deposition has the potential to increase the proportion of hexagonal organometallic rings.

Reply: We have performed this experiment. Figure 2 below indicates that thermal deposition has the potential to increase the proportion of hexagonal organometallic rings. We also add the thermal treatment in Figure 2a, and the sentence:

“From Figure S2c and S2d, it is clear that the proportion of non-hexagon monomers in the 2D networks has increased at RT. This means thermal deposition (at 365 K) has the potential to increase the proportion of hexagonal organometallic rings.”

Partial Figure 2. Crystallization of 2D organometallic networks on Cu(111) after Se doping. (a) Overview and (b) close-up STM images showing the ordered 2D organometallic networks fabricated by co-deposition of *m*TBPT and selenium on Cu(111) at 365 K. “L” and “R” domains of the formed networks, deviating $15 \pm 3^\circ$ from the $[0\bar{1}1]$ direction of the substrate.

Figure S2. Crystallization from random phase at RT by Se doping on Cu(111). (a) Large-scale and (b) close-up STM images showing the random phase formed by *m*TBPT on Cu(111) at RT. (c) Large-scale and (d) close-up STM images indicating the transformation from the random phase to 2D ordered crystalline phase after Se doping, still with some non-hexagonal pores. (e) Large-scale and (f) close-up STM images showing ordered 2D MOFs with few non-hexagonal pores after annealing at RT overnight (for 12 h). STM parameters: constant current, (a) $U = -1$ V, $I = 100$ pA; (b) $U = 500$ mV, $I = 100$ pA; (c) $U = 2$ V, $I = 10$ pA; (d) $U = 2$ V, $I = 1$ nA; (e, f) $U = 1$ V, $I = 100$ pA.

4) *What are the brighter and larger dots at the edge of the network in Fig. 2d, as opposed to the darker dots (Br adatoms)?*

Reply: The brighter and larger dots are C–Cu...Se interactions and small copper selenide domains, and the dark dots are Br adatoms as shown below in Figure R1.

Figure R1. (a) The current image and (b) corresponding nc-AFM frequency shift image of L-network acquired at the same position show the periphery interactions. Scanning parameters: (a) constant current mode with $U = 1$ V, $I = 100$ pA. (b) constant height mode using a CO-terminated tip at a bias voltage of 0 V, images were recorded at the tip offset $\Delta z = -240$ pm with respect to an STM set point ($U = 1$ V, $I = 100$ pA).

Considering that there is no such interaction at the periphery molecules before Se doping (surface stabilized radicals as shown in Figure S2b), we deduce that the interactions are due to Se. According to the higher-resolution STM and nc-AFM images shown below in Figure R2, we assign the brighter and larger dots at the edge of the network to C–Cu...Se interactions (white ellipses) and small copper selenide domains (white trapezoids). Detailed discussion about the C–Cu...Se interactions are provided in the Reply to Question 10 including not only experimental STM and nc-AFM results, but also the theoretical interaction region indicator (IRI) analysis.

Figure R2. (a) STM image of the ordered 2D metal organic nanostructure after mixing the *m*TBPT precursor with Se at 365 K. (b) current image and (c) nc-AFM frequency shift image acquired simultaneously of the same region in panel (a). Tunneling parameters: (a) constant current, $U = 1$ V,

$I = 100$ pA; (b, c) constant height mode with CO-functionalized tip at a bias voltage of 0 V, recorded at a tip offsets $\Delta z = -250$ pm with respect to an STM set point ($U = 1$ V, $I = 100$ pA).

5) *In 146 on page 8. At higher temperature, molecules have higher mobility on the surface, which is more conducive to the self-healing of 2D OM networks, and achieve a more thermodynamically stable structure. Why is it inconsistent with the conclusion of this work?*

Reply: We thank the reviewer for pointing out this issue and performed additional experiments. The revised manuscript includes:

“From Figures S2c and S2d, it is clear that the proportion of non-hexagon monomers in the 2D networks is increased at RT. Thermal deposition (at 365 K) increases the proportion of hexagonal organometallic rings. At higher temperature, molecules have higher mobility on the surface, which is more conducive to the self-healing of 2D OM networks, and achieve a more thermodynamically stable structure. In view of kinetics, mild annealing for sufficient time (i.e. RT for 12 h) could also increase the proportion of hexagon pores in the formed network as illustrated in Figure S2e and S2f.”

6) *In 154 on page 8. Is the diffusion barrier of copper selenide the primary factor hindering the formation of a large-scale network, or is it the insufficient coverage of the precursors?*

Reply: We performed further STM experiments with a higher coverage of precursors. As shown below, with increasing the coverage to 0.70 ML *m*TBPT doped with 0.015 ML Se, the 2D OM network grows bigger, but the structural homogeneity is restricted by domain boundaries with the coexistence of R and L chirality (Figure S3g). Hence, the diffusion barrier of copper selenide does not appear to be the primary factor hindering the formation of a large-scale network.

Figure S3g. Large-scale STM image showing 2D networks with higher coverage (0.70 ML *m*TBPT and 0.015 ML Se). STM parameters: constant current, $U = 1$ V, $I = 50$ pA.

7) *In the penultimate paragraph on page 8, the authors pointed that due to the limited diffusion energy barrier of copper selenide, it is impossible to generate a large two-dimensional network. However, the last paragraph on page 8 noted that Se atoms have a low migration energy barrier*

on Cu(111) and can spread rapidly, thus regulating product selectivity. This is contradictory, and the author should be consistent.

Reply: The copper selenide refers to the trapezoid domains as shown below in Figure R2, while the Se atoms refers to the single Se atoms. A single Se atom has low diffusion energy barrier of 0.107 eV according to the previous reference [ChemPhysChem 2016, 17 (14), 2137–2145]. However the copper selenide domains do not easily diffuse on Cu(111) surface since we could clearly resolved the trapezoid domains in STM images.

To avoid the misleading sentences, we deleted the discussion about the copper selenide and the single Se atom.

Figure R2. (a) STM image of the ordered 2D metal organic nanostructure after mixing the *m*TBPT precursor with Se at 365 K. (b) current image and (c) nc-AFM frequency shift image acquired simultaneously of the same region in panel (a). Tunneling parameters: (a) constant current, $U = 1$ V, $I = 100$ pA; (b, c) constant height mode with CO-functionalized tip at a bias voltage of 0 V, recorded at a tip offsets $\Delta z = -250$ pm with respect to an STM set point ($U = 1$ V, $I = 100$ pA).

8) *Why does the dI/dV spectrum located on the molecule have a peak at about +1.0 V (red curve).*

Reply: The peak at about +1.0 V (red curve) of the dI/dV spectrum was resolved in the unoccupied region, and we tentatively assign it to the conduction band edge (CBE).

In order to explore the origin of the STS state, we calculated the density of states (DOS) for the 2D OM network in gas phase by DFT. Figure S5a (also Figure 2h) displays the spin-polarized DOS of free-standing 2D OM network. Surprisingly, the asymmetric DOS for spin-up and spin-down components indicates a spin polarization of the 2D OM network, although the incorporated Cu atoms in the chains are not magnetically active, similar to our previous studies [Chem. Mat. 2022, 34, 1770–1777; J. Am. Chem. Soc. 2023, 145, 6203–6209].

The corresponding projected DOS (PDOS) on the Cu, C and N elements of the 2D OM network was also calculated to show individual contributions, where the Cu and C orbitals contribute to the highest occupied molecular orbitals (HOMO) and the C and N orbitals contribute to the lowest unoccupied molecular orbitals (LUMO) of the 2D OM network (Figure S5b).

The comparison of the measured data on the surface (Figure 2i) with the calculated results in gas phase reveals the influence of the Cu(111) surface on the electronic properties of the 2D OM network. The metal surface transfers electrons to the network as an electron donor, which is accompanied by a shift of all the bands downward in energy by approximate 2 eV with respect to the Fermi level, which is similar to Au⁻ and Ag⁻-based OM networks [Nanoscale, 2018, 10, 3769–3776; ACS Nano 2020, 14, 16887–16896]. No more details were observed from dI/dV spectra recorded across the *m*TBPT–Cu–*m*TBPT bridge, which could be attributed to the dominant DOS from the copper substrate [2D Mater. 2016, 3 (4), 045002; Nano Lett. 2020, 20 (2), 963–970].

Figure S5. DFT calculated electronic properties of the ordered organometallic network in gas phase. (a) Spin-polarized DOS (density of states), (b) projected density of state (PDOS), and (c) the corresponding DFT model of the ordered organometallic network in gas phase. Black and red lines in (a) represent the spin-up and spin-down contributions, respectively. Pink, gray and blue in (b) represent the Cu, C and N contributions, respectively.

9) In 215 on page 12. The data “0.12 eV” here is incorrect, the value should be 0.08 eV.

Reply: We thank the reviewer for his careful reading and have rewritten the sentences and paragraphs about Figure 4 with the revised DFT model.

10) The author needs to provide further characterization to more directly illustrate the existence of C–Cu...Se intermediates.

Reply: We performed further STM and nc-AFM experiments of the 2D *m*TBPT–Cu networks, especially on the periphery of the network, with the aim to find valuable clues that can shed light on the existence of C–Cu...Se intermediates.

As illustrated in Figure S6 (also shown below), terminal interactions to the periphery molecules were indicated by white ellipses. Considering that there is no such interaction at the periphery molecules before Se doping (surface stabilized radicals as shown in Figure S2b), we infer that the interaction is due to Se. Combined with the high-resolution STM and nc-AFM images shown below, we assign them (depicted by white ellipses) to C–Cu...Se interactions as highlighted in Figure S6e.

Figure S6. Periphery interactions C–Cu...Se of the network. (a) The current image and (b) corresponding nc-AFM frequency shift image of R-network acquired at the same position show the periphery interactions. (c) The current image and (d) corresponding nc-AFM frequency shift image of L-network acquired at the same position show the periphery interactions. White ellipses indicated the terminal interactions to the periphery molecules. (e) High-resolution of nc-AFM frequency shift image superimposed with the model indicates the C–Cu...Se interaction of the network periphery. Scanning parameters: (a, c) constant current mode with $U = 1$ V, $I = 100$ pA. (b, d, e) constant height mode using a CO-terminated tip at a bias voltage of 0 V, images were recorded at the tip offset $\Delta z = -250$ pm with respect to an STM set point ($U = 1$ V, $I = 100$ pA).

Interaction region indicator (IRI) analysis on the C–Cu...Se moiety was also performed to investigate the intermolecular interactions. To illustrate the nature of interactions between Se adatoms and C–Cu species, we used a finite C–Cu moiety with one Se atom terminated as a model system in the IRI analysis. IRI is a new real-space function that can clearly reveal both chemical bonds and weak interactions in chemical systems [Chem. Methods 2021, 1, 231–239]. As illustrated in the IRI isosurface in Figure S7 (shown in the reply to Reviewer 1), both covalent bond and weak interaction regions are nicely revealed by the blue and green isosurfaces, respectively, according to the standard coloring method. This suggests that there is nonnegligible interaction between the Se adatom and the C–Cu species.

11) Regarding the DFT calculations of the mechanism on topology selectivity, it would be one-sided and meaningless to solely analyze the energy advantage of the monomer. The interaction between molecules and substrate, and the migration of structures on the surface, etc., play crucial roles in determining the formation of the 2D networks. This is particularly true for Cu(111) substrate, where the interaction is notably strong, and the energy barrier for molecule migration on the surface cannot be overlooked.

Reply: We thank the reviewer for pointing out this issue. Additional DFT calculations were performed on energy barriers for molecule migration on the Cu(111) surface.

In the migration of the C_s -Cu moiety, as illustrated in Figure S8, the moiety has the same orientation in the initial and the final states. Along one principal axis (indicated by red arrows) the migration was calculated to be 0.68 eV. While along another principal axis (indicated by green arrows), the sliding diffusion was calculated to be 0.29 eV. Notably, migration along the third principal axis is, by symmetry, equivalent to the second (green) direction. Importantly, in both cases of the more stable initial and final states, the Cu adatoms were located in the more stable fcc sites, while in the transition states, the Cu adatoms were close to the less stable bridge sites. From a chemical point of view, the molecular moiety is nearly physisorbed on the surface (with distances between the triazine core and the substrate of more than 3 Å). Thus, compared with the molecule, the chemisorbed Cu adatoms in the C_s -Cu moiety play a more important role in the energy diagram.

Figure S8. DFT-calculated energy diagrams for sliding diffusion of the C_s -Cu moiety on Cu(111), where the top and side views of the paths are depicted in the left panel for two diffusion directions (indicated by red and green arrows respectively). The structural models are given for the initial, transition and final states, respectively. Brown, white, gray, blue, and pink balls represent C, H, N, Cu substrate, and Cu adatoms, respectively.

Similarly, in the migration shown in Figure S9, the C_{3h} -Cu moiety also has the same orientation in the initial and the final states. Along the first principal axis (indicated by red arrows), the sliding diffusion barrier was calculated to be 0.48 eV, while along the second principal axis (indicated by green arrows) the sliding diffusion barrier was calculated to be 0.24 eV, respectively. Due to the alignment of the moiety, migration along the third principal axis is equivalent to the first, red, direction.

Figure S9. DFT-calculated energy diagrams for sliding diffusion of the C_{3h} -Cu moiety on Cu(111), where the top and side views of the paths are depicted in the left panel for two diffusion directions (indicated by red and green arrows respectively). The structural models are given for the initial, transition and final states, respectively. Brown, white, gray, blue, and pink balls represent C, H, N, Cu substrate, and Cu adatoms, respectively.

In addition to the migration barriers, the rotation barriers of C_s -Cu and C_{3h} -Cu moieties on Cu(111) were also calculated. Figure S10 illustrates the rotation process where the C_s -Cu or C_{3h} -Cu moieties rotate around the fixed triazine core. The rotation is important in order for two organometallic moieties to properly align before combination. For the C_s -Cu moiety, shown in the upper panel of Figure S10, the moiety in the final state is rotated clockwise by 120° around a normal to the surface going centered at triazine core, compared to the initial state. For the C_{3h} -Cu moiety, depicted in the lower panel of Figure S10, the final state is similarly rotated clockwise by 120° compared to the initial state. The energy barriers are calculated to be 0.25 eV and 0.30 eV for C_s -Cu and C_{3h} -Cu moieties, respectively, indicating that the C_s -Cu and C_{3h} -Cu moieties can change their directions on the Cu(111) surface.

Hence, while the reviewer is correct in pointing out that the interaction on Cu(111) is notably strong, our calculations suggest that chemisorbed Cu adatoms in the C_s -Cu moiety play a more important role in the energy diagram. More importantly, neither the sliding diffusion nor the rotation diffusion processes are rate-limiting steps at room temperature.

Figure S10. DFT-calculated energy diagrams for rotation processes of the C_5 -Cu (upper panel) and C_{3h} -Cu (lower panel) moieties on Cu(111). The structural models are given for the initial, transition and final states, respectively. Brown, white, gray, blue, and pink balls represent C, H, N, Cu substrate, and Cu adatoms, respectively.

12) In the last paragraph on page 12. The analysis provided here lacks justification, because the formation of periodic structures is not a prerequisite for 2D OM networks on surface. Previous research has demonstrated that similar energy of the conformations of a precursor can cause a loss of coupling selectivity in 2D structures, leading to disorder (Chem. Commun. 2010, 46, 7157-7159; Nat. Mater. 2020, 19, 874–880)

Reply: We thank the reviewer for pointing out this issue. To better understand the mechanism for the high topology selectivity of the specific C_{3h} conformers out of a conformationally flexible precursor by Se doping, nudged elastic band (NEB) calculations in the absence or presence of Se have been employed to model the transformation between C_5 -Cu and C_{3h} -Cu moieties on Cu(111).

As shown in the revised Figure 4a (also shown previously), the energy barrier of C_5 transformed to C_{3h} without Se was calculated to be 0.52 eV, which can be overcome at RT. Considering the similar energy of C_5 and C_{3h} conformers, the random organometallics were formed with no obvious selectivity.

With Se doping, the transformation from C_5 to C_{3h} is a two-barrier process, where configuration with Se standing beside the phenyl lobe constitutes a shallow intermediate state and the configurations with Se connecting to the C_5 -Cu and C_{3h} -Cu constitute initial and final states. The energy barrier of C_5 transformed to C_{3h} with Se was calculated to be 0.60 eV from the initial state to the intermediate state, and 0.48 eV for the second barrier from the intermediate state to the final state (Figure 4b). Conversely, the reverse energy barrier of C_{3h} transformed to C_5 with Se was calculated to be 0.99 eV with no energetically favorable intermediate state. As previously discussed using the Arrhenius equation, the Boltzmann ratio $r \gg 1$ in quite a wide range of reaction temperature. Large values of r indicates that the transformation from C_5 to C_{3h}

is preferred rather than the reverse transformation from C_{3h} to C_s after Se doping, which explains the high selectivity of C_{3h} conformers after doping.

Reviewer #3 (Remarks to the Author):

In this manuscript, the authors used Se-doping strategy to achieve the topology selectivity of a conformationally flexible precursor and construct an ordered 2D metal organic network on Cu(111) surface. The main conclusion is supported by high-resolution STM and nc-AFM investigations. Furthermore, a plausible mechanism is proposed based on the experimental examinations as well as DFT calculation.

The findings introduce an important approach to the growing field of on-surface chemistry, which opens new avenues towards the synthesis of chiral low-dimensional materials. The manuscript is overall well written, is based on high-quality experimental data, an insightful analysis and touches a timely topic.

Prior to publication, a series of points deserves further attention:

1. The annealing temperatures should be presented in the corresponding STM images (Figures 1 and 2).

Reply: We thank the reviewer for the positive comments and constructive suggestions. The annealing temperatures are presented in the revised Figure 1 and 2 as shown below.

Figure 1. Intact conformers and the random phase formed by *mTBPT* on Cu(111). (a) Overview and (b) close-up STM images showing isolated monomers and self-assembled domains with limited order after deposition of *mTBPT* on Cu(111) at 90 K. The close-packed directions of Cu(111) are marked by white arrows. (c, d) High-resolution STM images indicating the intact conformers R-*C_s*-, L-*C_s*-, R-*C_{3h}*-, and L-*C_{3h}*-*mTBPT* together with their schematic models. (e) Overview and (f) close-up STM images after annealing to 365 K, showing the random phase of C–Cu–C organometallics formed by fully debrominated *mTBPT* precursors and Cu adatoms. (g) Statistical analysis of 1200 molecules indicating no obvious selectivity of *mTBPT* conformers on Cu(111). STM parameters: constant current, (a, e, f) $U = 1$ V, $I = 100$ pA; (b-d) $U = -1$ V, $I = 100$ pA.

Partial Figure 2. Crystallization of 2D organometallic networks on Cu(111) after Se doping. (a) Overview and (b) close-up STM images showing the ordered 2D organometallic networks fabricated by co-deposition of *m*TBPT and selenium on Cu(111) at 365 K. “L” and “R” domains of the formed networks, deviating $15\pm 3^\circ$ from the $[0\bar{1}1]$ direction of the substrate.

2. The chemical structure of organometallic network can be fitted in Figure 2 (partially fitted). That may be reader-friendly.

Reply: The chemical structure of organometallic network is partially fitted in the revised Figure 2 for friendly reading.

Partial Figure 2. (d) High-resolution of “L” domain with molecular model partially fitted. The green line from A to B is a line scan profile with the measured dimension of the *m*TBPT–Cu–*m*TBPT species shown in (e). The unit cell is indicated by black dashed rhombus. (f) High-resolution of “R” domain with molecular model partially fitted. Red and black triangles contour the brighter (I) and dim (II) molecules, and blue circles contour the interlinked Cu adatoms. The dashed triangle frames a bright molecule.

3. The STS measurements of network are strongly affected by the surface state of Cu(111). It is helpful to show the gas-phase DOS of this network to have a better understanding of its electronic properties.

Reply: Gas-phase density of states (DOS) of this network has been calculated to better understand its electronic properties. The spin-polarized DOS of free-standing 2D network was

indicated by Figure S5a (also Figure 2h). Surprisingly, the asymmetric DOS for spin-up and spin-down components indicates a spin polarization of the 2D network, although the incorporated Cu atoms in the chains are not magnetically active, similar to our previous studies [Chem. Mat. 2022, 34, 1770–1777; J. Am. Chem. Soc. 2023, 145, 6203–6209].

The corresponding projected DOS (PDOS) on the Cu, C and N elements of the 2D OM network was also calculated to show individual contributions, where the Cu and C orbitals contribute to the highest occupied molecular orbitals (HOMO) and the C and N orbitals contribute to the lowest unoccupied molecular orbitals (LUMO) of the 2D OM network (Figure S5b).

The comparison of the measured data on the surface (Figure 2i) with the calculated results in gas phase reveals the influence of the Cu(111) surface on the electronic properties of the 2D OM network. The metal surface transfers electrons to the network as an electron donor, which is accompanied by a shift of all the bands downward in energy by approximate 2 eV with respect to the Fermi level, which is similar to Au⁻ and Ag⁻-involved OM networks [Nanoscale, 2018, 10, 3769–3776; ACS Nano 2020, 14, 16887–16896]. No more details were observed from dI/dV spectra recorded across the *m*TBPT–Cu–*m*TBPT bridge, which could be attributed to the dominant DOS from the copper substrate [2D Mater. 2016, 3 (4), 045002; Nano Lett. 2020, 20 (2), 963–970].

Figure S5. DFT calculated electronic properties of the ordered organometallic network in gas phase. (a) Spin-polarized density of states (DOS), (b) projected density of state (PDOS), and (c) the corresponding DFT model of the ordered organometallic network in gas phase. Black and red lines in (a) represent the spin-up and spin-down contributions, respectively. Pink, gray and blue in (b) represent the Cu, C and N contributions, respectively.

*4. The DFT calculations partially explained the transformation of random phase to ordered network. The topology selectivity of conformations is the key point of this research, this crucial issue requires a more rigorous treatment. The transformation from C_s to C_{3h} conformer by flipping one of *m*-bromophenyl groups was confirmed by author's experimental data. What is the flipping process, and the role of Se atom? The Se atoms improve the reversibility of C-Cu-C bond, just like the role of I, Br, and Cl atoms during the on-surface synthesis?*

Reply: To better understand the mechanism for the high topology selectivity by Se doping, we performed extensive DFT calculations especially the energy barriers of C_s transformed to C_{3h} with and without Se. As shown in the revised Figure 4a (shown in the reply to Reviewer 1), the

energy barrier of C_s transformed to C_{3h} without Se was calculated to be 0.52 eV, which can be overcome at RT. Considering the similar energy of C_s and C_{3h} conformers, random organometallics were formed with no obvious selectivity.

With Se doping, the transformation from C_s to C_{3h} is a two-barrier process, where configuration with Se standing beside the phenyl lobe constitutes a shallow intermediate state and the configurations with Se connecting to the C_s -Cu and C_{3h} -Cu constitute initial and final states. The energy barrier of C_s transformed to C_{3h} with Se was calculated to be 0.60 eV from the initial state to the intermediate state, and 0.48 eV for the second barrier from the intermediate state to the final state (Figure 4b). Conversely, the reverse energy barrier of C_{3h} transformed to C_s with Se was calculated to be 0.99 eV with no energetically favorable intermediate state. This is the same argument as explained previously. We thus answer the reviewer's question of the role of Se atom by this energy transition state calculation. We agree that possibly the Se atoms improve the reversibility of C-Cu-C bond, just like the role of I, Br, and Cl atoms during the on-surface synthesis.

Reviewer #4:

In this manuscript (MS), Cai et al. present an intriguing way of controlling the conformationally of organometallic (OM) structures made of triphenylazine (mTBPT) precursor on Cu(111) using selenium. A combination of STM, nc-AFM and DFT is used to identify the chemistry of the OM structure, establish a statistical analysis of the different conformers and rationalize the effect of selenium. After deposition of mTBPT on Cu(111), two different phases and its chiral fashions are found: R-Cs, L-Cs, R-C3h and L-C3h. The statistical analysis, as expected, shows that there is no selectivity for one of these phases and their almost equally present. Upon introduction of Se (independently of the deposition sequence), the authors claim that the presence of R/L-Cs conformers is significantly reduced and only R/L-C3h is found. This transition and its perspective are very interesting. Tuning the symmetry of a system and selecting a specific among various options is of great interest. The explanation of this effect and the identification of the underpinning mechanism would enable the translation of this work on to different systems.

However, the experimental demonstration and reasons why the presence of an amount of Se atoms on Cu(111) can produce this transition are unclear. The four phases expected from the deposition of mTBPT result from its conformational flexibility and its characteristic prochiral properties that mainly play during the deposition process. To have the transition, all the molecules in a Cs form would have to transform into the C3h form by reversing one of the m-bromophenyl groups, as rightly reported on page 12. This transition seems very unlikely from an energetical point of view due to the barrier to overcome; the cited references 10 (not relevant), 44 (refers to the gas phase), 45 (not relevant) provide no clear explanation.

The current provided DFT simulations rely on the assumption that the symmetry of the two-dimensional network can be ducted from the orientation of single molecule calculations and their positioning with respect to the bridging Cu atoms. The calculation suggest that C3h symmetry is the more stable than Cs by 0.08 eV in the absence of Se. This trend is the same with the consideration of Se (0.14 eV). These calculations suggest that both assemblies should be present on the surface and they do not show the selectivity. I would expect a much larger enthalpic loss for Cs and I suspect that it can be revealed from periodic 2D network simulations. In addition, the experiments suggest that the conformer information of the molecule changes when adding Se as the Cs conformers vanish and have to transform to C3h.

Therefore, I would only recommend accepting this MS after thoroughly reviewing the theoretical elaboration and after obtaining further experimental results showing, on the same surface area, the transformation of the Cs form into the C3h form due to the close presence of Se atoms.

Reply: According to the reviewer's suggestion, we performed additional STM experiments and DFT calculations to investigate the mechanism of Se doping, as previously described. For the experimental part, we performed further STM experiments with different coverage of Se and different ratios of mTBPT to Se. All samples were thermally treated at 365 K for 2 hours after co-deposition of mTBPT and Se.

When 0.005 ML Se and 0.28 ML *m*TBPT were mixed on Cu(111), there coexisted C_s and C_{3h} conformers, forming organometallic chains and networks as shown in Figure S3a and S3b (shown in reply to Reviewer 1). After increasing the coverage of Se to 0.01 ML (Figure S3c and S3d), only the C_{3h} conformer was observed. That is to say, the amount of Se is related to the stabilization of the C_{3h} conformer. Therefore, we deduce that the C_{3h} conformer could be stabilized with the doped Se atoms. Considering the low diffusion barrier of Se atoms on Cu(111) (0.107 eV) [ChemPhysChem 2016, 17 (14), 2137–2145], it is plausible that the quickly diffusing Se atoms play an important role in regulating the precursor conformers and topology selectivity. More details about how the Se atoms influence the energy diagram are provided in the DFT discussion.

In order to further demonstrate this deduction, the coverage of *m*TBPT was increased to 0.70 ML (Figure S3e and S3f), and the network grew bigger. The chain structure composed of the C_s conformer was observed again for the insufficient Se. With increasing Se to 0.015 ML (Figure S3g and S3h), the C_s conformer was observed less.

For the theoretical part, NEB calculations in the absence or presence of Se have been employed to model the transformation between C_s -Cu and C_{3h} -Cu moieties on Cu(111). As shown in the revised Figure 4 (shown in reply to Reviewer 1), the energy barrier of C_s transformed to C_{3h} without Se was calculated to be 0.52 eV, which can be overcome at RT. Considering similar energy of C_s and C_{3h} conformers, random organometallics were formed with no obvious selectivity.

As previously discussed, with Se doping, the transformation from C_s to C_{3h} is a two-barrier process, where configuration with Se standing beside the phenyl lobe constitutes a shallow intermediate state and the configurations with Se connecting to the C_s -Cu and C_{3h} -Cu constitute the initial and final states. The energy barrier of C_s transformed to C_{3h} with Se was calculated to be 0.60 eV from the initial state to the intermediate state, and 0.48 eV for the second barrier from the intermediate state to the final state (Figure 4b). Whereas the energy barrier of C_{3h} transformed to C_s with Se was calculated to be 0.99 eV with no energetically favorable intermediate state.

These calculations suggest that both assemblies should be present on the surface and they do not show the selectivity, except in the presence of Se. In addition, the experiments suggest that the conformer information of the molecule changes when adding Se as the C_s conformers vanish and have to transform to C_{3h} . The additional experimental results and theoretical elaboration should support the transformation of the C_s form into the C_{3h} form due to the close presence of Se atoms.

In the following additional remarks and comments.

General remarks:

The quality of the Figures is lacking. In Figure 1f, the structural model is very pixelated. However, understanding the different conformers is key to appreciate the work. Moreover, Figure 4 should be redesigned; there is too much information in the Figure.

All experiments are conducted well, but some noise and scratches are present on the images (e.g. Figure 1d) that could be cleaned up.

Reply: We have performed additional STM experiments and DFT calculations on the energy barriers for the conformer transformation. The revised Figure 1 and Figure 4 are depicted below showing less noise and scratches in Figure 1, and less pixilation in Figure 4.

Figure 1. Intact conformers and the random phase formed by *m*TBPT on Cu(111). (a) Overview and (b) close-up STM images showing isolated monomers and self-assembled domains with limited order after deposition of *m*TBPT on Cu(111) at 90 K. The close-packed directions of Cu(111) are marked by white arrows. (c, d) High-resolution STM images indicating the intact conformers R-*C_s*-, L-*C_s*-, R-*C_{3h}*-, and L-*C_{3h}*-*m*TBPT together with their schematic models. (e) Overview and (f) close-up STM images after annealing to 365 K, showing the random phase of C–Cu–C organometallics formed by fully debrominated *m*TBPT precursors and Cu adatoms. (g) Statistical analysis of 1200 molecules indicating no obvious selectivity of *m*TBPT conformers on Cu(111). STM parameters: constant current, (a, e, f) $U = 1$ V, $I = 100$ pA; (b-d) $U = -1$ V, $I = 100$ pA.

Figure 4. Energy diagrams for the transformation from C_s to C_{3h} (a) without and (b) with Se doping. The structural models are given for the initial, transition, intermediate and final states, respectively. The energy scale is not linear. Energies are given in units of eV. Brown, white, gray, green, blue, and pink balls represent C, H, N, Se, Cu substrate, and Cu adatoms, respectively.

I suggest deleting the term “wide” relating to the temperature used (room temperature to 365 K). The amount of Se atoms must be indicated carefully: “low coverage” is not clear enough. I suggest modifying Scheme 1 to clearly include the four species, as Scheme 1 of Ref. 45.

Reply: We have deleted the term “wide” as the temperature is from RT to 365K.

And the amount of Se or coverage of Se is defined as 1 ML Se is defined as 1 Se atom per surface Cu atom, which is in accordance with the previous study [ChemPhysChem 2016, 17, 2137–2145]. We also revised it in the Methods and Materials section.

The term “low coverage” is now quantified, e.g. in Abstract: “by low coverage (0.01 ML) Se doping.”

We also modify Scheme 1 to clearly include four species (as in Scheme 1 of Ref. 45) according to the reviewer’s suggestion, as shown below:

Scheme 1. 2,4,6-Tris(3-bromophenyl)-1,3,5-triazine (*m*TBPT) Conformers with C_s and C_{3h} Symmetries, Right-Handed (R) and Left-Handed (L) Chirality. Reaction pathway of *m*TBPT on Cu(111) toward random organometallics (OMs) and crystalline 2D metal organic nanostructures after selenium doping.

Comments:

On the transition of the conformer:

Figure 1 demonstrates that all conformers and their chiral versions (*R/L-C_{3h}* and *R/L-C_s*) exist almost equally. The addition of Se to this configuration transforms the monomers to change to C_{3h} exclusively (Figure 2). This means that either the phenyl rotates with respect to the triazine core or that the radical migrates along the C-C backbone. The authors have to investigate the catalytic effect of Se. Nudged elastic band calculations in the absence or presence of Se can be employed to model this rotation. If the activation barrier drastically decreases, it could be the key to understand the experimental results.

Reply: We have performed extensive experiments with different coverage of Se and different ratios of *m*TBPT to Se, in order to investigate the mechanism of Se doping (see more details on the discussion of Figure S3 (shown above)).

Nudged elastic band calculations in the absence or presence of Se were also employed to model the transformation between the C_s and C_{3h} conformers (see more details and the discussion of the revised Figure 4 as shown above).

DFT calculations in Figure 2 of the adsorbed 2D OM phase and the unit cell:

The authors write that experimentally the lattice parameters are: $a = b = 23.2 \text{ \AA}$. However, the DFT simulated unit cell has the lattice parameters: $a = b = 22.28 \text{ \AA}$. While I do not believe that

the change of surface orientation between DFT and experiment is too important, the lattice strain is 4.13 %, which is quite large. This strong compressive strain on the network will likely make the periodic system wavy to release the strain. The authors use this wavy nature to explain the different contrasts between the monomers in the unit cell. However, it seems to me more like a DFT artifact due to the strong compression. Can the authors improve the simulation and reduce the strain on OM structure? Could the authors provide the lattice parameters of the network in gas-phase and compare them to the current simulations?

Reply: After careful repeated measurements of more STM images on the 2D OM network, we revised the experimental lattice parameters as: $a = b = 22.8 \pm 0.8 \text{ \AA}$.

After comparison of models with different lattices, this ($a = b = 22.28 \text{ \AA}$) is the most stable one. The orientation of the 2D OM network is therefore in agreement with the experimental one, and there is unlikely to be much strain.

The lattice parameters of the network in gas-phase is: $a = b = 22.33 \text{ \AA}$.

On the alternating contrast difference:

In Figure 3, the authors show highly resolved images of the OM structure using nc-AFM. The dimmer molecule of the unit cell seems very flat. In contrast the triazine core of the other molecule appears bright. The same is observed in the STM images. Beyond a possible wavy nature of the OM network, which comes at an enthalpic cost and should make it less stable, are there other possibilities to recreate this contrast? Could the triazine core act as a Se trap and the Se intercalates below as a single atom?

Reply: We have considered a single Se atom intercalating between the triazine core and the Cu(111) substrate (see below in Figure R3). After relaxation, the Se atom must simultaneously move adjacent to the H atoms with no energy barrier, which seems energetically unlikely. The brightness difference of the triazine core of the neighboring molecules is derived from their height difference of 0.22 \AA .

Figure R3. Models with a single Se atom intercalating between the triazine core and the Cu(111) substrate before and after relaxation.

DFT simulation of intact molecules Page 11 line 202-210:

The authors have carried out simulations of the two different conformers before dehalogenation in the absence and presence of Se. “These results suggest that the presence of Se atoms can effectively modulate the energy landscape of the system, favoring the formation of specific conformers”. While the statement is true, I do not see how this relates to the work and how it improves the rationalization of the mechanism. The precursors immediately dehalogenate upon adsorption. Therefore, the experiment will not resemble the simulations. Moreover, the adsorption site of the Se atom in Figure S5d is quite odd. There should be some repulsion between the Se and the halogens, which could provide an enthalpic loss and could be the reason for the 0.11 eV difference.

Reply: We have deleted the previous Figure S5, and performed additional theoretical calculations with dehalogenated molecules to revise the mechanism (see more details in the revised Figure 4).

DFT simulation of dehalogenated molecules with ad atoms in different adsorption sites without Se atoms (Figure 4):

The authors have carried out DFT calculations to compute the energy difference between different adsorption geometries. “By careful comparison between the calculated 2D MOF in Figure 2g-h and the C_{3h}-Cu moieties in Figure 4a-f, it was observed that the network is composed of C_{3h}-I and C_{3h}-II moieties, as highlighted by the red and black rectangle” Could the authors visualize this? Moreover, the argumentation “that the network is composed of C_{3h}-I and C_{3h}-II moieties” relies on the heavily strained system, which makes it questionable. Can the authors please comment how they extract relevant information from the single molecule adsorption geometries for the 2D OM network? The calculation as of now suggest that almost all configurations are valid as their energies are quite comparable <0.2 eV.

Reply: In the original discussion, by careful observation of the calculated building block of the ordered 2D OM network in original Figure 2h (also shown below in Figure R4), we extracted the C_{3h}-I and C_{3h}-II moieties, considered their orientations and adsorption sites, and relaxed the models by DFT for simplification.

We agree with the reviewer that the original calculations are not convincing enough. Therefore, we performed additional theoretical calculations, especially the NEB calculations in the absence or presence of Se, in order to model the transformation between the C_s and C_{3h} conformers (see more details in the revised Figure 4). Figure 4 has been revised and the discussion has been rewritten. We are no longer using the original C_{3h}-I and C_{3h}-II models.

Figure R4. Original Figure 2(h): Top-view of the DFT relaxed building block with the unit cell indicated by black dashed rhombus. Original Figure 4(a) and 4(b): DFT-relaxed models of C_{3h}-I and C_{3h}-II moieties before Se doping. Gray, white, blue, brown, and pink balls represent C, H, N, Cu substrate, and Cu adatoms, respectively.

DFT simulation of dehalogenated molecules with ad atoms in different adsorption sites with Se atoms (Figure 4):

The authors compared the aforementioned trend of energies without Se atoms to similar simulations with Se atoms. They state “This regulation resulted in the favouring of specific conformers (C_{3h}-I and C_{3h}-II in Figure 4m and 4n), exhibiting lower energies compared to other configurations and indicating increased stability, thus finally facilitated the formation of the desired 2D MOFs with high topology selectivity.” The question is how would the structure in Figure 4m,n form the 2D OM network, if the Se atoms are placed in such a way that incoming monomers cannot connect to the Cu atom because the Se atom blocks it. Most of the proposed structures are having this issue.

Reply: We have calculated the energy barrier for breaking the interaction between C-Cu and the Se atom in the C-Cu...Se terminal as shown in Figure R5. In view of kinetics, the C-Cu terminals prefer to connect with Se atoms, in agreement with our experimental results. But the low energy barrier (0.57 eV) still provides the Se atoms sufficient mobility so that bigger 2D OM networks can form at RT~365 K.

Figure R5. DFT-calculated energy diagram for breaking the interaction between C–Cu and the Se atom in the C–Cu...Se terminal. The structural models are given for the initial, transition and final states, respectively. Brown, white, gray, green, blue, and pink balls represent C, H, N, Se, Cu substrate, and Cu adatoms, respectively.

I believe the authors should focus on explaining how the transformation from C_s to exclusively C_{3h} occurs. Se atoms seem to catalyse this transition and it should be reflected in a kinetic point of view through the activation barriers of the dihedral rotation. The presented picture on the thermodynamic site seems to evaluate the effect of the Se on the adsorption of the monomer on Cu(111) (some structures are less planar than others, e.g. Figure 4n compared to Figure 4u,p) and not the formation of the 2D network. I could imagine that the Se decouples the organic backbone from the surface and prevents hybridization, which would facilitate rotation as the molecule-substrate interactions are weaker.

Reply: We thank the reviewer for the constructive suggestions on the theoretical calculations. According to the comments, NEB calculations in the absence or presence of Se were also employed to model the transformation between the C_s and C_{3h} conformers (see more details and the discussion of the revised Figure 4).

Computational methods section 4):

The authors do not state the construction of the surface slaps. Moreover, the $2 \times 2 \times 1$ k-point grid seems quite low to make comparison in the 100 meV range as the total energy values may still fluctuate with increasing k-points.

Reply: The Cu(111) surface was modeled as a three-layer slab vertically separated by a vacuum region of 15 Å. For the transformation calculations we used a $p(7 \times 7)$ surface unit cell in the absence of Se, whereas a $p(9 \times 9)$ surface unit cell was used in the presence of Se, together with a $5 \times 5 \times 1$ k-point grid in both cases. We tested different k-points in this system by calculating binding energies of an intact C_{3h} molecule on Cu(111) with different k-points, and finally the $5 \times 5 \times 1$ k-point grid was adopted in the revised manuscript.

Table. Binding energies of an intact C_{3h} molecule on Cu(111) with different k-points.

	C_{3h} on Cu(111)/eV	Cu(111)/eV	C_{3h} in gas/eV	Binding energy/eV
kpts 1x1x1	-815.530	-548.200	-263.294	-4.036
kpts 2x2x1	-813.437	-546.235	-263.296	-3.906
kpts 2x2x2	-813.408	-546.223	-263.295	-3.890
kpts 3x3x1	-815.742	-548.491	-263.298	-3.953
kpts 3x3x4	-815.712	-548.476	-263.295	-3.942
kpts 4x4x1	-815.181	-547.967	-263.296	-3.918
kpts 5x5x1	-815.678	-548.497	-263.296	-3.885
kpts 6x6x1	-815.397	-548.216	-263.296	-3.885

Reviewer #5 (Remarks to the Author):

I co-reviewed this manuscript with one of the reviewers who provided the listed reports as part of the Nature Communications initiative to facilitate training in peer review and appropriate recognition for co-reviewers.

Reply: We also thank the reviewer for the contribution.

REVIEWER COMMENTS

Reviewer #1 (Remarks to the Author):

In this revised version, the authors have revised the manuscript according to my comments and the others concerns. The revisions, including additional theoretic IRI and DFT analysis and supplementary STM results, have significantly improved the ms quality and fully supported the major conclusion. Therefore, I recommend the manuscript for publication as it stands.

Reviewer #2 (Remarks to the Author):

The authors have made significant revision on this manuscript. The revised manuscript reads much better now. Overall, I am satisfied with their revision and point-to-point response. Therefore, I recommend to accept it for publication in Nature Commun. now.

Reviewer #4 (Remarks to the Author):

See attached file.

Referee report on the reviewed version of the manuscript “Topology Selectivity of a Conformationally Flexible Precursor by Selenium Doping” submitted by Cai et al. for publication in Nature Communications.

The authors made great efforts to improve the manuscript (MS), that is appreciated. The experimental comments have been well addressed and the improved scheme and Figures are more readable.

However, the rationalisation of the experimental observations with the presented DFT and NEB calculations is difficult to follow. I agree that the calculations in the absence of selenium show that there is no selectivity. Figure 4b shows the NEB calculation including the effect of selenium atoms. The reaction path shows that there is an intermediate state between the conformers. The authors argue that asymmetric reaction barriers are responsible for the high selectivity. I agree with the reasoning, but I think there should also be another reaction pathway that is likely to occur. The pathway presented in the MS is divided into two sections: first, the Se atom decouples from the Cu atom with an endothermic gain of 0.51 eV (IntS) and then the phenyl rotates by reattaching itself to the Cu atom with an equal energy compared to the initial state. The asymmetry between the first step of decoupling and the second step of simultaneous rotation and coupling, I believe, is the cause of the asymmetric reaction barriers. The authors show in Figure R5 that a similar intermediate state can be achieved by the C3h configuration by first decoupling Se from Cu with a similar endothermic gain (0.55 eV, FS in Fig. R5). From this result, I believe it can exist another intermediate state for which the barrier for phenyl rotation could be reduced. The reaction barriers of this process should then clearly determine the effect of the Se atom.

Therefore, I recommend modelling the reaction by first decoupling the Se atoms as in Figure 4b (IS->IntS), then rotating the phenyl Cu complex and then coupling the Se atom.

I continue to believe that this manuscript is of high importance to the field and still recommend its publication after addressing this concern.

Reviewer #5 (Remarks to the Author):

1) Reviewer #4 (Remarks to the Author):

The authors made great efforts to improve the manuscript (MS), that is appreciated. The experimental comments have been well addressed and the improved scheme and Figures are more readable.

However, the rationalisation of the experimental observations with the presented DFT and NEB calculations is difficult to follow. I agree that the calculations in the absence of selenium show that there is no selectivity. Figure 4b shows the NEB calculation including the effect of selenium atoms. The reaction path shows that there is an intermediate state between the conformers. The authors argue that asymmetric reaction barriers are responsible for the high selectivity. I agree with the reasoning, but I think there should also be another reaction pathway that is likely to occur. The pathway presented in the MS is divided into two sections: first, the Se atom decouples from the Cu atom with an endothermic gain of 0.51 eV (IntS) and then the phenyl rotates by reattaching itself to the Cu atom with an equal energy compared to the initial state. The asymmetry between the first step of decoupling and the second step of simultaneous rotation and coupling, I believe, is the cause of the asymmetric reaction barriers. The authors show in Figure R5 that a similar intermediate state can be achieved by the C_{3h} configuration by first decoupling Se from Cu with a similar endothermic gain (0.55 eV, FS in Fig. R5). From this result, I believe it can exist another intermediate state for which the barrier for phenyl rotation could be reduced. The reaction barriers of this process should then clearly determine the effect of the Se atom.

Therefore, I recommend modelling the reaction by first decoupling the Se atoms as in Figure 4b (IS->IntS), then rotating the phenyl Cu complex and then coupling the Se atom.

I continue to believe that this manuscript is of high importance to the field and still recommend its publication after addressing this concern.

Reply: We thank the reviewer for the positive comments and pointing out this issue. After careful consideration of our experimental observations, we performed extensive calculations including the effect of selenium atoms to rationalize our experimental observations with the DFT calculations, and revised the manuscript as follows:

After Se doping, the Se atoms locate beside the phenyl lobe which is about to rotate (IS). During the rotation, the adjacent Se atoms help to decouple the organic backbone from the surface and prevents hybridization, which would facilitate rotation as the molecule-substrate interactions are weaker. The rotation barrier from initial state via transition state to intermediate state is calculated to be 0.32 eV, followed by a shallow barrier for the Se atom couple to the C_{3h} -Cu species (FS). The low rotation barrier of 0.32 eV elucidates the ease with which C_s -Cu species can rotate to C_{3h} -Cu species in the presence of Se on Cu(111) surface. The subsequent shallow barrier suggests that Se can readily bind with C_{3h} -Cu species to form the stable final state. The entire pathway is exothermic by 0.58 eV (FS). This explains the formation of the inner C_{3h} -Cu- C_{3h} interlinked network as well as the peripheral C_{3h} -Cu...Se interactions.

We have also calculated the energy barrier for breaking the interaction between C_{3h} -Cu and Se atom in the C_{3h} -Cu...Se terminal as shown in Figure S8 (also previous Figure R5). In

view of kinetics, the C_{3h} -Cu terminals prefer to connect with Se atoms, in agreement with our experimental results (periphery C_{3h} -Cu...Se interactions around the network). But the low energy barrier (0.57 eV) still provides the Se atoms sufficient mobility so that bigger 2D OM networks can form at RT~365 K. The reverse shallow barrier shows how easily the Se could couple to C_{3h} -Cu with an exothermic loss of 0.55 eV. The Arrhenius equation is defined as:

$$v = A \exp(-\Delta E/k_B T),$$

where the energy barrier ΔE is defined as the energy difference between the TS and IS, and k_B is Boltzmann's constant. The prefactor A is normally assigned the rule-of-thumb value of 10^{13} s^{-1} . To study the influence of the calculated energy barriers on the reaction kinetics, we define the Boltzmann ratio as:

$$r = \exp(-(\Delta E_1 - \Delta E_2)/k_B T)$$

where ΔE_1 equals to 0.32 eV and ΔE_2 equals to ~0.58 eV as shown in Figure 4b. As a result, the Boltzmann ratio $r \gg 1$ in quite a wide range of reaction temperature. Large values of r indicates that the transformation from C_s to C_{3h} is preferred rather than the reverse transformation from C_{3h} to C_s after Se doping, which accounts for the high selectivity of C_{3h} conformers after doping.

The decoupling of Se atom from the C_s -Cu species (possible process before the IS) could be excluded for the following reason: The overall energy of the C_{3h} -Cu network with peripheral C_{3h} -Cu...Se interactions was more stable compared with that of the C_s -Cu chain structure with terminal C_s -Cu...Se interactions. Single C_{3h} -Cu...Se or C_s -Cu...Se interaction is equal in energy for their intrinsic C-Cu...Se interaction. However, the number differs in different organometallic structures. Imaging there are equal numbers of C_{3h} or C_s precursors constituting the organometallic network or chain respectively, ideally the C_s -Cu chain structure owns only two terminals available for further C_s -Cu...Se coupling while the C_{3h} -Cu network possesses a lot more peripheral terminals for further C_{3h} -Cu...Se coupling to stabilize the final structure. This quantitative difference explains that the overall energy for the C_{3h} -Cu network was more favorable compared with that for C_s -Cu chain structure when coupling to the Se atoms.

In experiments when we deposited the Se on the random organometallics at low temperature (~ 200 K), the selenium (or copper selenide) was also observed to locate beside the C_s -Cu species, in addition to form C_s -Cu...Se terminals (Figure S9). Therefore, we excluded the decoupling of Se atom from the C_s -Cu species for simplify, start with the configuration with Se beside the phenyl lobe, and try to rationalize our experimental observations with the DFT calculations by two steps: first, rotation of C_s -Cu species to C_{3h} -Cu species with the help of Se atoms locating beside the lobe and then followed by the coupling of Se atom attaching to the C_{3h} -Cu species to stabilize the final structure.

We hope these further calculations and discussions could rationalize our experimental observations with the DFT calculations.

Revised Figure 4. Energy diagrams for the transformation from C_s to C_{3h} (a) without and (b) with Se doping. The structural models are given for the initial, transition, intermediate and final states (IS, TS, IntS and FS), respectively. The energy scale is not linear. Energies are given in units of eV. Brown, white, gray, green, blue, and pink balls represent C, H, N, Se, Cu substrate, and Cu adatoms, respectively.

Figure S8 (also Previous Figure R5). DFT-calculated energy diagram for breaking the interaction between C–Cu and the Se atom in the C–Cu...Se terminal. The structural models are given for the initial, transition and final states, respectively. Brown, white, gray, green, blue, and pink balls represent C, H, N, Se, Cu substrate, and Cu adatoms, respectively.

Figure S9. The STM image showing deposition of Se at low temperature (~ 200 K) on the Cu(111) surface pre-covered with random organometallics. The copper selenide outlined by white trapezoid is located beside a C_s –Cu moiety. STM parameters: constant current, $U = 1$ V, $I = 100$ pA.

REVIEWERS' COMMENTS

Reviewer #4 (Remarks to the Author):

In this revised version, the authors have revised the manuscript according to the reviewers' comments. I recommend the publication in this form.

Reviewer #5 (Remarks to the Author):
